# Coupling of oceanic carbon and nitrogen facilitates spatially resolved quantitative reconstruction of nitrate inventories

Nicolaas Glock[1], Zeynep Erdem [2], Klaus Wallmann[1], Christopher J. Somes [1], Volker Liebetrau[1], Joachim Schönfeld[1], Stanislav Gorb[3] & Anton Eisenhauer[1]

Anthropogenic impacts are perturbing the global nitrogen cycle via warming effects and pollutant sources such as chemical fertilizers and burning of fossil fuels. Understanding controls on past nitrogen inventories might improve predictions for future global biogeochemical cycling. Here we show the quantitative reconstruction of deglacial bottom water nitrate concentrations from intermediate depths of the Peruvian upwelling region, using foraminiferal pore density. Deglacial nitrate concentrations correlate strongly with downcore $\delta^{13}C$, consistent with modern water column observations in the intermediate Pacific, facilitating the use of $\delta^{13}C$ records as a paleo-nitrate-proxy at intermediate depths and suggesting that the carbon and nitrogen cycles were closely coupled throughout the last deglaciation in the Peruvian upwelling region. Combining the pore density and intermediate Pacific $\delta^{13}C$ records shows an elevated nitrate inventory of >10% during the Last Glacial Maximum relative to the Holocene, consistent with a $\delta^{13}C$-based and $\delta^{15}N$-based 3D ocean biogeochemical model and previous box modeling studies.

[1] GEOMAR Helmholtz Centre for Ocean Research Kiel, Wischhofstrasse 1-3, Kiel 24148, Germany. [2] NIOZ Royal Netherlands Institute for Sea Research and Utrecht University, Landsdiep 4, 1797 SZ 't Horntje, Texel, The Netherlands. [3] Zoological Institute: Functional Morphology and Biomechanics Kiel University, Am Botanischen Garten 9, Kiel 24118, Germany. Correspondence and requests for materials should be addressed to N.G. (email: nglock@geomar.de)

Nitrogen (N) is a fundamental component of amino acids and thus, essential for all living organisms[1]. The increasing use of chemical fertilizers to provide food for a growing global population and the burning of fossil fuels lead to a severe rise of fixed nitrogen in the biosphere[2]. Nitrate ($NO_3^-$) is one of the main limiting nutrients in the modern ocean[3] and nitrate fertilization is considered to contribute to the ongoing ocean deoxygenation[4,5]. A strong climate sensitivity has been predicted for the global $NO_3^-$ inventory and feedbacks on climate by the coupling of the biogeochemical carbon (C) and nitrogen (N) cycles through the biological pump[1]. A quantitative reconstruction of past reactive N inventories and feedbacks on other biogeochemical cycles throughout time might help us to predict scenarios for the future. Nevertheless, despite different estimates from numerical models, quantitative paleo-records for past reactive N budgets in the oceans are not yet available.

The main source for bioavailable N in the modern ocean is $N_2$-fixation, performed by cyanobacteria[6,7], while the main loss of mineralized N results from denitrification and anaerobic ammonium oxidation (Anammox) in oxygen deficient zones (ODZs) in the sediments as well as in the water column[7–9]. Estimates of past inventories of reactive N species are mainly based on geological records of the fractionation between the N isotopes $^{15}N$ and $^{14}N$ (given as $\delta^{15}N$ in ‰) in bulk sedimentary organic matter ($\delta^{15}N_{bulk}$). The complexity of various Ncycle processes influencing $\delta^{15}N$ (e.g. $N_2$ fixation, sedimentary or water column denitrification, $NO_3^-$ utilization, remineralization and nitrification) complicates a quantitative reconstruction of the N budget based on $\delta^{15}N$ alone.

Models have used global $\delta^{15}N_{bulk}$ records for estimating changes in the past N budget. Box modeling studies[10,11] agree that the inventory of reactive N was likely elevated during cold phases mainly due to a reduction of denitrification in the water column and seafloor sediments, related to enhanced $O_2$ solubility in colder seawater and decreased area of shelf sediments from lower sea level, respectively. Additionally, enhanced $N_2$ fixation from atmospheric iron deposition has been proposed[8,12]. Estimated changes from box models based on $\delta^{15}N_{bulk}$ range from 5 to 100%[7,10,11] increase of reactive N during glacials as compared to interglacials. Another box model study, which is not based on $\delta^{15}N_{bulk}$[13], predicts changes in the global oceanic nutrient budgets due to changes in sea-level, dust deposition, and ocean circulation. This study estimates an increase in dissolved N (DN) of ~16% during the late Holocene compared to the Last Glacial Maximum. This is generally consistent with a study representing glacial nitrogen cycling constrained by isotopes in a 3D global ocean biogeochemical model considering LGM boundary conditions that predicts a glacial $N_{bio}$ increase between 6.5 and 22%[7].

A main focus of our study is the reconstruction of past $NO_3^-$ concentrations ($[NO_3^-]$) using the pore density of benthic foraminifera. Foraminifera are one of the rare examples of eukaryotes which are able to use $NO_3^-$ as an electron acceptor when oxygen is depleted within their habitats and play an important role in the oceanic benthic nitrogen cycle[14–16]. The pore density in the shells of Bolivina spissa is significantly correlated to the $[NO_3^-]$ in their habitats because the pores facilitate the uptake of electron acceptors for respiration[17]. A comprehensive review about the functionality of pores in benthic foraminifera can be found in ref. [18]. The functionality of pores in Foraminifera ranges from gas exchange for the uptake of electron acceptors and the release of metabolic waste products like $CO_2$[19] to the uptake of dissolved organic material[20]. Foraminifera from oxygen depleted environments typically show an increased porosity[21] and often a clustering of mitochondria under the pores[19,22]. Several recent studies describe the influence of oxygen availability on foraminiferal pore characteristics[23–25]. While some species adapt

their porosity by changing the size of their pores[25], other species are adapting the numbers of pores (pore density) in their tests[17,23,24].

Benthic Foraminifera from oxygen depleted environments have recently been shown to use $NO_3^-$ as electron acceptor[14,15]. At least one species, B. spissa, from the Peruvian ODZ, adapts its pore density to the availability of $NO_3^-$ in its habitat[17]. A comparison of 232 measurements of the pore density in B. spissa to the bottom water nitrate concentrations ($[NO_3^-]_{BW}$ from 8 different sampling locations at the Peruvian continental margin revealed a significant linear relationship between both parameters. Another species from the Peruvian ODZ, Bolivina seminuda, has been shown to have a high affinity to $NO_3^-$ availability[26]. The tests of B. seminuda are highly porous[17]. Every species of the genus Bolivina which has been analyzed so far, including B. seminuda, has the ability to denitrify[15,27], which implies that denitrification is a common strategy of Bolivinidae for survival under oxygen depleted conditions. This makes species from this genus in particular candidates for paleo $NO_3^-$ reconstruction by analyses of pore characteristics as an empirical proxy.

We determined the pore density of the benthic foraminiferal species B. spissa as a quantitative paleoproxy for $[NO_3^-]$ in intermediate waters (1250 m) at the Peruvian continental margin over the last deglaciation. The foraminiferal pore density is providing a tool to reconstruct past $[NO_3^-]$ in a high lateral and temporal resolution allowing to test model predictions. A comparison of the reconstructed $[NO_3^-]$ to the stable carbon isotope ratio ($\delta^{13}C$) in our sedimentary record shows the same correlation as in intermediate depths of the modern Pacific, enabling us to reconstruct regional differences in deglacial $[NO_3^-]$. A first analysis of deglacial $\delta^{13}C$ records reveals the same trend in deglacial $[NO_3^-]$ change as reconstructed by the pore density and predicted by the different model studies.

## Results

**Deglacial changes in the oceanic reactive N inventory.** We reconstructed bottom water $NO_3^-$ concentrations ($[NO_3^-]_{BW}$) using sediment core M77/2 52-2 (5°29′S; 81°27′W; 1250 m) from the Peruvian continental margin over the last deglaciation. Past $[NO_3^-]_{BW}$ was reconstructed using the pore density of the benthic foraminiferal species B. spissa (Fig. 1a, b; Supplementary Table 1) following the method published in ref. [18]. The pore densities of 819 specimens were analyzed for this record to provide a statistically robust dataset in a sufficient temporal resolution (Fig. 1a). We distinguished between five different time intervals including the Last Glacial Maximum (LGM; 22–17 kyr BP), Heinrich Stadial 1 (H1; 17–15 kyr BP), Antarctic Cold Reversal (ACR; 15–12 kyr BP), Early Holocene (EH; 11.7–8.2 kyr BP) and Middle to Late Holocene (MLH; 8–0 kyr BP). The lowest pore densities (highest $[NO_3^-]_{BW}$) occurred during the LGM. This difference is highly significant compared to all other individual time intervals ($P < 0.001$; $N = 136$; two-sided heteroscedastic Student's T-test). The highest pore densities, and thus lowest $[NO_3^-]_{BW}$, have been found for the MLH. This difference is also highly significant compared to all other time intervals ($P < 0.001$; $N = 353$).

A comparison with continuous transient global box model simulations covering the last deglaciation[10,11,13] provided evidence that the $NO_3^-$ inventory at this location is driven by fluctuations of the global reactive N-inventory. A plot which compares relative changes in the global reactive N budget over the last deglaciation from the different modeling approaches with our quantitative $[NO_3^-]_{BW}$ record is shown in Fig. 1c. The pore density derived $NO_3^-$ inventory during the LGM was elevated compared to the Holocene which corroborates estimations from

previous biogeochemical model studies[7,10,11,13] although it also reveals fluctuations in $[NO_3^-]_{BW}$ in a much higher temporal resolution. A 50-100% higher reactive N inventory is suggested for the LGM by another box model study by Eugster et al.[11] and thus probably overestimates this change by one order of magnitude according to our reconstruction.

The $\delta^{15}N_{bulk}$ record of our sediment core (Fig. 1d) showed trends similar to other records from the Eastern Tropical South Pacific (ETSP) and the Eastern Tropical North Pacific (ETNP)[10,28]. It depicted the typical maximum of $\delta^{15}N_{bulk}$ in these regions during the last deglaciation, which was caused by an acceleration in water column denitrification relative to the LGM[10,28]. The increase in benthic denitrification at the shallow shelf due to sea level rise and increased shelf area was slower than the increase of denitrification in the water column. The balancing between enhanced denitrification in the water column and sedimentary denitrification by $N_2$ fixation, which introduces low $\delta^{15}N$ into the ocean, at the onset of the Holocene leads to a subsequent reduction in $\delta^{15}N_{sed.org}$[29].

**Deglacial coupling of $\delta^{13}C$ and $NO_3^-$ in intermediate depths.** A comparison between the reconstructed $[NO_3^-]_{BW}$ and $\delta^{13}C$ measured on *Uvigerina peregrina* in the same sediment core

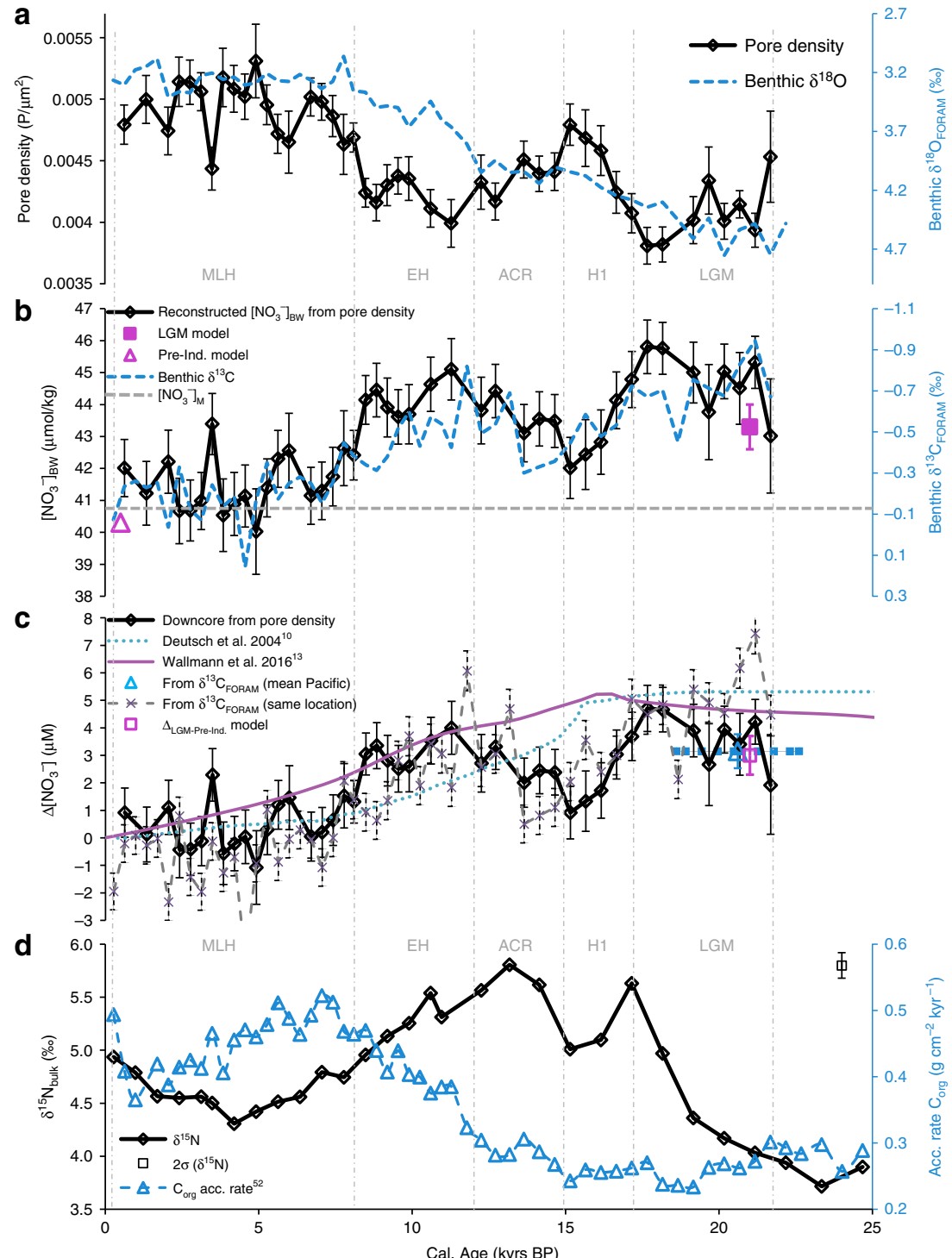

($\delta^{13}C_{FORAM}$, Fig. 1b) showed a strong coupling starting from the LGM and persisting over deglaciation until the Late Holocene. The mean $\delta^{13}C$ signature of dissolved inorganic carbon ($\delta^{13}C_{DIC}$) in seawater is controlled by the balance between terrestrial and marine carbon sources and sinks[13,30–32], while the spatial distribution of $\delta^{13}C_{DIC}$ is mainly controlled by photosynthesis, respiration, and the ventilation and mixing between different water masses[30–33]. Autotrophic organisms preferably take up the lighter isotope $^{12}C$ during photosynthesis. Thus, surface water masses have more positive $\delta^{13}C_{DIC}$ and are depleted in DIC, since $^{12}C$-carbon is preferably exported as organic matter. In intermediate to deep water masses organic matter is readily remineralized by respiration, which leads to an increase in DIC and a decrease of $\delta^{13}C_{DIC}$ within these water masses. An increase in photosynthesis leads to a higher export productivity and thus a stronger gradient in $\delta^{13}C_{DIC}$ between surface and deep water masses established through the biological carbon pump.

Since photosynthesis and respiration both influence the distribution of major nutrients in the ocean, there is an inverse relationship between $\delta^{13}C_{DIC}$ and $[NO_3^-]$ and $[PO_4^{3-}]$ in the modern ocean, with a stronger correlation to $[NO_3^-]$ than $[PO_4^{3-}]$[33]. The distributions of $[NO_3^-]$ and $\delta^{13}C_{DIC}$ in water masses of the modern Pacific, taken from the GLODAPv2 database[34], are shown in Fig. 2a, b. Both distributions are similar since the main processes affecting $\delta^{13}C$ also affect the $NO_3^-$ distribution. Surface water masses show high $\delta^{13}C_{DIC}$ and low $[NO_3^-]$ through primary productivity, while the intermediate to deep water masses show low $\delta^{13}C_{DIC}$ and higher $[NO_3^-]$ through remineralization of exported organic matter. All these processes define the endmembers of $\delta^{13}C_{DIC}$ and $[NO_3^-]$ in different water masses and thus the mixing processes between different water masses follow the same trend.

A comparison of the correlation of downcore $[NO_3^-]_{BW}$ and $\delta^{13}C_{FORAM}$, and the correlation of dissolved $NO_3^-$ and $\delta^{13}C_{DIC}$ in intermediate water depths (700–2000 m) of the recent Pacific[34], is shown in Fig. 2c. Both linear regressions were highly significant ($P < 0.001$) and neither slopes nor intercepts significantly differed from each other (Slope: $P = 0.15$; Intercept: $P = 0.13$). Thus, $[NO_3^-]_{BW}$ and $\delta^{13}C$ in our downcore record showed basically the same correlation over the last 22 kyrs as $[NO_3^-]$ and $\delta^{13}C$ of DIC in intermediate water depths of the modern Pacific. We propose that the linear regression between $[NO_3^-]$ and $\delta^{13}C_{DIC}$ (eq. 1 and eq. 2) can be used to quantitatively reconstruct

past $[NO_3^-]$.

$$\delta^{13}C_{DIC} = -0.093(\pm 0.001) \cdot [NO_3^-] + 3.568(\pm 0.038) \quad (1)$$

Alternatively solved for $[NO_3^-]$:

$$[NO_3^-] = -\frac{(\delta^{13}C_{DIC} - 3.568(\pm 0.038))}{(0.093(\pm 0.001))} \quad (2)$$

**3D Biogeochemical model on deglacial $\delta^{13}C_{DIC}$-$[NO_3^-]$ coupling.** The distribution of $\delta^{13}C$ and $[NO_3^-]$ has been modeled for the modern ocean, the pre-industrial Holocene and the LGM (Fig. 3) using a coupled 3D ocean circulation-biogeochemical isotope model. The model system used here is an improved version of Somes et al.[7] by including the carbon isotope cycling following Schmittner and Somes[35,36] and optimizing LGM iron deposition patterns to better reproduce $\delta^{15}N_{bulk}$ observations (see Supplementary Figure 1). The modeling results indicated no significant difference in the relationship of the $\delta^{13}C_{DIC}$-$[NO_3^-]$ correlation in the deep intermediate Pacific at our core location (i.e. $[NO_3^-]_{BW}$ μM; Supplementary Figure 2) during the different climatic time intervals. This supported our comparison of the M77/2-52-2 sediment record to the modern $\delta^{13}C_{DIC}$-$[NO_3^-]$ distribution. The $[NO_3^-]_{BW}$ reconstruction using our pore density proxy during the LGM and MLH at our sampling location corresponded well to our independent global biogeochemical model based on sedimentary $\delta^{15}N_{bulk}$ records. The predictions of our global 3D biogeochemical model for the sampling location of M77/2 52-2 are shown in Fig. 1b for the LGM and the pre-industrial Holocene. The best prediction from this model of 7.4‰ (uncertainty range 2.7–11‰; Supplementary Table 2), was generally consistent with the relative offset in the nitrate inventory between the LGM and MLH of ~10‰ from our pore density record. It has to be noted that the model predicted that the increase to the global $[NO_3^-]$ inventory was 1.5 μM larger than at our core location.

**Intermediate Pacific $[NO_3^-]$ records by the use of $\delta^{13}C_{Foram}$.** The reconstructed relative $[NO_3^-]_{BW}$ changes from the pore density of *B. spissa* and another $[NO_3^-]_{BW}$ reconstruction based on the $\delta^{13}C_{FORAM}$ record on *U. peregrina* and equation 2 are showing the same trends and magnitude (Fig. 1c). The

**Fig. 1** Quantitative $NO_3^-$ reconstruction and additional proxy records for sediment core M77/2 52-2 and comparison to different modeled $NO_3^-$ budgets. **a** Pore density of *Bolivina spissa* and $\delta^{18}O_{FORAM}$ measured on *Uvigerina peregrina* (core M77/2 52-2). Error bars represent the standard error of the mean (1 SEM). Single data points represent mean pore density of 7–22 specimens (see Supplementary Table 1). **b** $[NO_3^-]_{BW}$ calculated from the pore density of *B. spissa* after equation 4 and inverse $\delta^{13}C_{FORAM}$ measured on *U. peregrina* (core M77/2 52-2). Error bars represent 1 SEM including a complete error propagation (see equations 5 and 6). Magenta symbols: Model predictions from our 3D global biogeochemical model based on $\delta^{15}N_{bulk}$ for the location of M77/2 52-2. Gray line: Modern $[NO_3^-]_M$ in the same water depth taken from the closest station available in the GLODAPv2 database[34] (Station see Methods). **c** Black: Relative changes of $[NO_3^-]_{BW}$ calculated from the pore density of *B. spissa* (core M77/2 52-2) compared to modern $[NO_3^-]$ (indicated in **b**). Error bars represent 1 SEM. Turquoise line: Modeled relative changes of global $[NO_3^-]$ based on global $\delta^{15}N_{bulk}$ records (modified after ref. [10]; model run for strong water column denitrification feedback)[10]. Magenta line: Relative changes of global dissolved inorganic nitrogen (DIN) predicted by the boxed earth system model from ref. [13]. Gray crosses: Record of relative $[NO_3^-]_{BW}$ change based on the $\delta^{13}C_{FORAM}$ measured on *U. peregrina* (core M77/2 52-2) using equation 2. Blue triangle: Relative change of $[NO_3^-]$ at the intermediate Pacific between LGM (19–23 kyrs BP) and Late Holocene (0–6 kyrs BP). Relative $[NO_3^-]$ change was also calculated after equation 2 using the offset of mean $\delta^{13}C_{FORAM}$ measured on *Cibicidoides* spp. between the two time intervals in 14 sediment records from the Pacific. Error bars represent 1 SEM. Data has been taken from Petersen et al.[37] and two additional references[75,76]. See Methods section for location details and local variability. Magenta square: Model predictions from our 3D global biogeochemical model based on $\delta^{15}N_{bulk}$ for the location of M77/2 52-2 ($\Delta_{LGM-Pre-Ind.}$: Offset between both time intervals from **b**). **d** Record of $\delta^{15}N_{bulk}$ and accumulation rates of organic matter[52] in M77/2 52-2. The error bar is representing the standard deviation (2σ) of $\delta^{15}N$ measurements on the reference standard (Acetanilide)

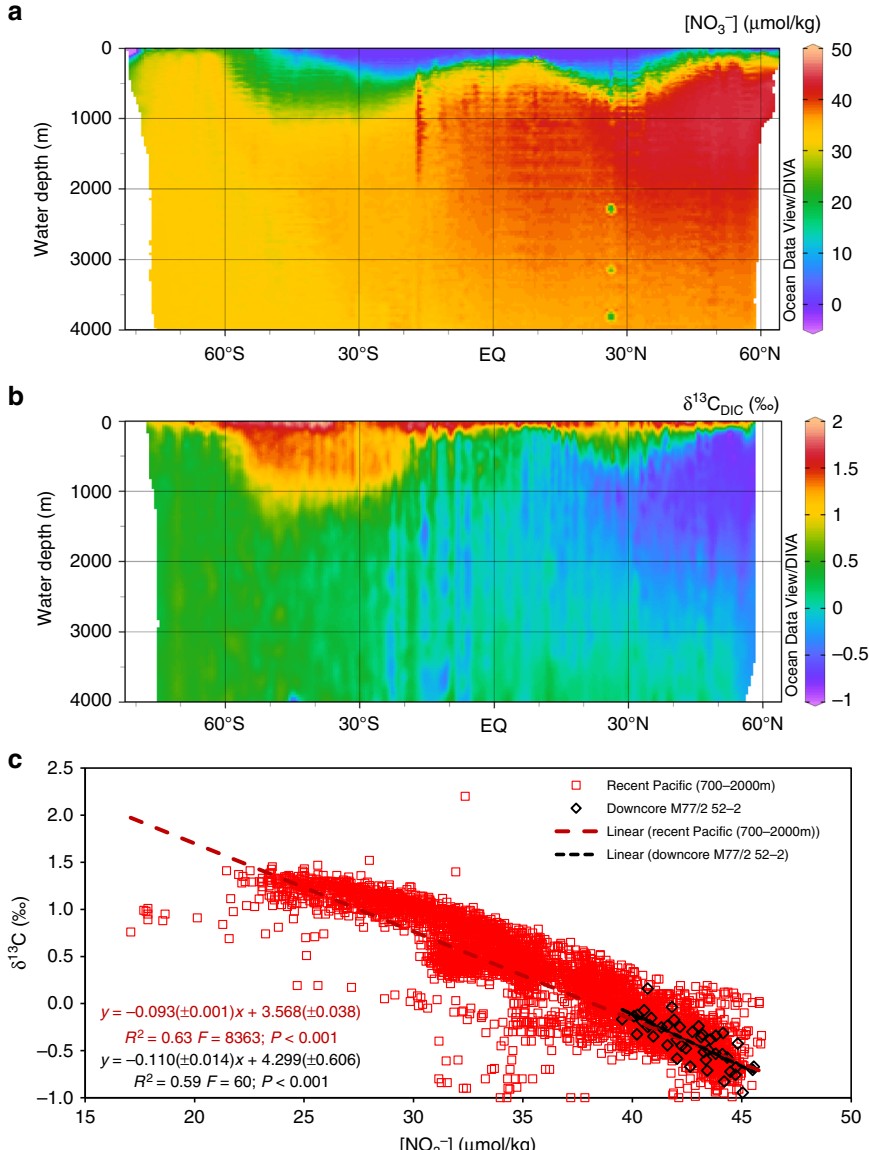

**Fig. 2** Distribution of $NO_3^-$ and $\delta^{13}C_{DIC}$ in the modern Pacific and $[NO_3^-]$-$\delta^{13}C_{DIC}$-coupling in the intermediate Pacific - modern and downcore. All data for the modern Pacific have been taken from the GLODAPv2 database[34]. **a** Distribution of $[NO_3^-]$ in the modern Pacific[34]. **b** Distribution of $\delta^{13}C$ of dissolved inorganic carbon (DIC; $\delta^{13}C_{DIC}$) in the modern Pacific[34]. The Ocean Data View software has been used to compile these plots[62]. **c** Correlation between $[NO_3^-]$ and $\delta^{13}C_{DIC}$ in intermediate water depths (700-2000 m) of the modern Pacific (red, $N = 4779$) and between $[NO_3^-]_{BW}$ and $\delta^{13}C_{FORAM}$ in the sediment record of M77/2 52-2 (black, $N = 44$). Both linear regressions neither differ significantly in slope ($P = 0.15$) nor in intercept ($P = 0.13$). Due to graphical reasons all $\delta^{13}C$ below $-1‰$ have been cut in this plot, although they were included into the fit. For a complete plot of all data points see Supplementary Figure 4B

$\delta^{13}C_{FORAM}$ based reconstruction is more noisy. This is probably due to the sample size and numbers of measurements for each data point. The pore density record averages individual measurements on a higher number of specimens ($N \sim 20$), while the $\delta^{13}C_{FORAM}$ record consists only of a single measurement on a bulk sample of a few specimens ($N \sim 5$). Another reason might be an influence of microhabitat preferences of *U. peregrina* on $\delta^{13}C_{FORAM}$ (Supplementary Note 1).

Globally, more than 400 records of $\delta^{13}C_{FORAM}$ measured on epifaunal *Cibicidoides* spp. are available[37]. From the compilation of downcore records we extracted all available $\delta^{13}C_{FORAM}$ records from Pacific intermediate water depths (700-2000 m, Supplementary Table 3). This provides the possibility to test if the

$\delta^{13}C_{DIC}$-$[NO_3^-]$-correlation might be also used at different sampling locations. Using the offset between mean $\delta^{13}C$ from the LGM (19-23 kyrs BP) to the late Holocene (0-6 kyrs BP), we calculated the average relative change of $[NO_3^-]$ between both time intervals. Once again, this result indicates that $[NO_3^-]$ was 3.0 ($\pm0.5$ 1 SEM; $N = 14$) μmol/kg higher during the LGM (Fig. 1c). These first tests indicate that $\delta^{13}C_{FORAM}$ from Pacific intermediate water depths might indeed be used to reconstruct deglacial $[NO_3^-]$ changes. An attempt to use these records to reconstruct regional differences in the intermediate Pacific is shown in Supplementary Figure 3 (see Methods section for details).

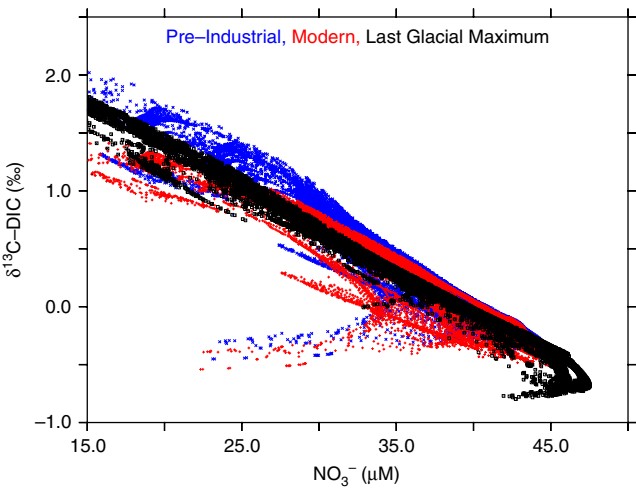

**Fig. 3** Model simulations of $[NO_3^-]$-$\delta^{13}C_{DIC}$ coupling in the intermediate Pacific for different time intervals. Correlation between $[NO_3^-]$ and $\delta^{13}C_{DIC}$ in intermediate water depths (700–2000 m) of the Pacific for the for the modern (red crosses; 1990–2010 average after accounting for decreased atmospheric $\delta^{13}C_{CO2}$), pre-industrial (blue x's) and LGM (black squares) from the 3D ocean biogeochemical isotope model

## Discussion

In this study, we show the application of a new quantitative $NO_3^-$ paleoproxy using pore density of the benthic foraminiferal species *B. spissa*. Furthermore, we propose that $\delta^{13}C_{FORAM}$ in benthic foraminifera from Pacific intermediate water depths is directly coupled to $[NO_3^-]$ (Fig. 1). A first comparison of different available $\delta^{13}C_{FORAM}$ records measured on tests of the epibenthic *Cibicidoides* from the Pacific at intermediate water depths show similar trends as the pore density record and global biogeochemical model predictions (Fig. 1c).

We found a distinct offset of $[NO_3^-]_{BW}$ between the LGM and the MLH (Fig. 1b, c). The depletion of reactive N during warm periods compared to glacial periods can be explained by lower denitrification activity during the glacials[1,7,10–12]. $N_2$ fixation may have also been stimulated by enhanced iron deposition[11–13], although $\delta^{15}N_{sed.org}$ records from the tropical North Pacific and Atlantic indicate reduced $N_2$ fixation during glacials[38,39] in response to reduced N loss, consistent with our 3D biogeochemical isotope model. Iron fertilization also led to additional export production and transport of remineralized $NO_3^-$ into the deep Southern Ocean waters. This resulted in a reduction of preformed $[NO_3^-]$ in Subantarctic Mode Waters (SAMW), which supply the tropical regions with preformed nutrients, affecting $NO_3^-$ limitation at lower latitudes[7].

A stronger stratification of Antarctic water masses due to decreased meridional overturning during the LGM probably supported the storage of remineralized nutrients in sluggish Antarctic Bottom Water and thus supported the decrease of preformed $NO_3^-$ during the LGM[40]. This led to a decreased transport of preformed $NO_3^-$ to the tropics limiting productivity, which reduces the volume of ODZs and thus denitrification[7]. Furthermore, the low sea level during the LGM led to a reduction of shelf seafloor area from 0 to 100 m water depths by 73%[13]. Shelf and hemipelagic sediments are the main contributors to sedimentary N loss processes[29,41] today. Both processes, water column denitrification in ODZs and sedimentary denitrification, the main sinks for reactive N, were dampened during the LGM compared to the MLH[28]. The most distinctive offset to the global model predictions appears during H1, when $[NO_3^-]_{BW}$ was depleted for ~4 kyrs (Fig. 1b). This offset most probably

represents local dynamics not accounted for in the coarse resolution of box model studies which are discussed in the Supplementary Note 2 together with local $O_2$ fluctuations and their possible influence on local $[NO_3^-]_{BW}$.

A comparison between the reconstructed $[NO_3^-]_{BW}$ and $\delta^{15}N_{bulk.}$ (Fig. 1b, d) in our sediment core shows a phase shift at the beginning of the last deglaciation (~18 kyr BP): High $[NO_3^-]_{BW}$ during the LGM corresponds to more isotopic light $\delta^{15}N_{bulk}$ while $[NO_3^-]_{BW}$ and $\delta^{15}N$ were in phase during the deglaciation and the Holocene. At first glance, this might appear contradicting since heavier $\delta^{15}N_{bulk}$ indicates higher water column denitrification, which would result in $NO_3^-$ depletion. However, sediment core M77/2 52-2 is located in intermediate water depths well below the most oxygen ($O_2$) depleted center of the ODZ near the thermocline. Deglacial water column denitrification mainly occurred in ODZs, and was probably stimulated by an enhanced supply of preformed nutrients that led to an increase in export production. As such, more organic N was transferred to intermediate water depths by the biological pump where it was decomposed to $NO_3^-$ and increased ambient $[NO_3^-]_{BW}$ despite the N loss at shallower water depths.

Our comparison of $\delta^{13}C_{FORAM}$ to the reconstructed $[NO_3^-]_{BW}$ using the pore density proxy show how closely the oceanic carbon and nitrogen cycle were coupled over the last glacial/interglacial cycle in the Pacific. $\delta^{13}C_{FORAM}$ has extensively been used as a proxy of paleoproductivity before but also as proxy for ventilation and oxygenation[42,43]. Nevertheless, our study shows that the ratio between $[NO_3^-]_{BW}$ and $\delta^{13}C_{FORAM}$ over the last 22 kyrs at our sampling location remained unchanged and implies the possibility that $\delta^{13}C_{FORAM}$ might also be used as a quantitative $NO_3^-$ proxy at intermediate water depths. Several locations of the modern Pacific show relatively low $\delta^{13}C_{DIC}$ values (Supplementary Figure 4). The positions of these locations mainly follow the distribution of anthropogenic $CO_2$ in the Pacific (Scupplementary Note 3 and Supplementary Figure 5). Since the deglacial correlation between $\delta^{13}C_{FORAM}$ and reconstructed $[NO_3^-]_{BW}$ is not influenced by anthropogenic $CO_2$, deviations from this correlation could even be used to trace anthropogenic $CO_2$ in the modern ocean.

A factor controlling the mean $\delta^{13}C_{DIC}$ in seawater is the exchange of atmospheric $CO_2$ with the ocean surface. A change in atmospheric $pCO_2$ would also mediate disequilibrium in the surface ocean. However, a recent study showed that this $pCO_2$ effect would cause a maximum $\delta^{13}C_{DIC}$ offset in subsurface waters of the Southern Ocean of ~0.2‰[44]. This deglacial offset is even smaller in other parts of the oceans and close to zero at our sampling location and thus cannot explain the changes of $\delta^{13}C_{Foram}$ in our downcore record. Despite this low deglacial offset in $\delta^{13}C_{DIC}$ by the $pCO_2$ effect the authors of named study caution to interpret $\delta^{13}C_{Foram}$ as a nutrient proxy. The $pCO_2$ effect might mask the influence of the biological pump on $\delta^{13}C_{DIC}$ if the $pCO_2$ gradient is very strong at times of high atmospheric $pCO_2$ such as during the early Cenozoic.

The fact that the correlation between $\delta^{13}C_{DIC}$ and $NO_3^-$ in intermediate water depths of the Pacific was stable over the last deglaciation is unexpected at a first glance. Indeed, the main processes which control the distribution of $\delta^{13}C_{DIC}$ and $NO_3^-$ in the oceans all influence both parameters as discussed above. However, the main factors controlling the oceanic $NO_3^-$ budget (e.g., denitrification and $N_2$ fixation) do not individually influence $\delta^{13}C_{DIC}$ in the same direction. It is possible that the main background driver controlling both processes is the deglacial change in sea level. The decreased area of continental shelves during the LGM in comparison to interglacial conditions led to a lower benthic denitrification and thus higher $[NO_3^-]$, and a lower burial rate of organic carbon and thus a lower mean

oceanic $\delta^{13}C$. Strong nitrogen cycle feedbacks are required to realistically model deglacial $\delta^{15}N$[10]. In this case, the main factor controlling the oceanic $NO_3^-$ budget would indeed be the change in benthic denitrification due to the extension of shelf seafloor. The mean oceanic $\delta^{15}N$ is controlled by the ratio of pelagic to benthic denitrification[28] and the balance from $N_2$ fixation. The decrease of $\delta^{15}N_{bulk}$ in the Eastern Tropical North and South Pacific starting ~12 kyr BP can indeed be modeled by increasing the ratio of benthic to pelagic denitrification[28] since benthic denitrification fractionates $\delta^{15}N$ much less than pelagic denitrification.

Another factor, controlling both $\delta^{13}C_{DIC}$ and $[NO_3^-]$ in different water masses is the ventilation and thus their reservoir age. Consistent evidence from different studies indicate a poorly ventilated deep Pacific during the LGM[45–47]. Data from the Eastern Equatorial Pacific (EEP) is very scarce, though, and shows some strong contrasts between different sampling locations and approaches[45,47–49]. Nevertheless, it is likely that deep water masses at the EEP were also poorly ventilated during the LGM. This older water mass would increase $[NO_3^-]$ and reduce $\delta^{13}C_{DIC}$ by remineralization, as well as reduce oxygen concentration ($[O_2]$) at these depths.

Contrarily, redox proxy records from the EEP indicate higher $[O_2]$ during the LGM at depths similar to our sampling location[50]. This is consistent with other redox proxy records from shallower depths in the Peruvian upwelling region, which indicated a less pronounced ODZ and lower primary productivity during the LGM[51]. Indeed, the accumulation rates of organic carbon (Acc. Rate. $C_{org}$) at our sampling site were lower during the LGM[52] which also indicates a lower primary productivity above this sampling site (Fig. 1d). The elevated $[O_2]$ during the LGM are in disagreement with poorly ventilated water masses and thus cannot directly explain the tendencies within our record due to local changes in water mass ventilation. This suggests that local changes to overlying productivity have a strong impact on $[O_2]_{BW}$, whereas $[NO_3^-]_{BW}$ is more influenced by the global $NO_3^-$ inventory that is determined by the large-scale balance between $N_2$ fixation and denitrification. It might well be, though, that total changes in the nutrient budget of the Pacific are partly related to an increased reservoir age of the deep water masses, related to decreased meridional overturning.

The comparison of our record to the other deglacial $\delta^{13}C_{FORAM}$ records is considered as evidence, that coupling of $[NO_3^-]$ and $\delta^{13}C_{DIC}$ was mainly controlled by the biological carbon pump at our sampling location in the intermediate Eastern Equatorial Pacific, and possibly at other regions of the intermediate Pacific. The situation might be different in the Atlantic Ocean, at greater depths or further back in Earth's history. Independent calibrations are thus substantial to extend the application of $\delta^{13}C_{Foram}$ as a quantitative $[NO_3^-]$ proxy. Furthermore, it is unlikely that this proxy would work at shallow depths where the $pCO_2$ effect might predominate the effect of the biological carbon pump.

Our 3D biogeochemical modeling results support that $[NO_3^-]$ at our sampling location records changes in the global budget (predicted at our location: $\Delta NO_3^- = 3.0\ \mu M$), but also is affected by iron fertilization at high latitudes. Iron fertiliation decreases preformed nutrients in SAMW and shallow Antarctic Intermediate Water (AAIW), where our core location exists, of the Pacific due to the transfer of more remineralized nutrients to the deep Pacific. This process is observationally constrained in the 3D model by direct comparison to $\delta^{15}N_{bulk}$ across the Southern Ocean (Supplementary Figure 1), which records changes to surface $[NO_3^-]$ utilization in response to dust deposition[7]. Sensitivity simulations associated with Southern Ocean iron fertilization uncertainties cause $[NO_3^-]$ changes at our location of

$\pm 0.7\ \mu M$ on top of the direct impact on global $[NO_3^-]$ (Supplementary Table 2). The increase to global $[NO_3^-]$ in the model was 1.5 $\mu M$ larger than bottom water $[NO_3^-]$ change at our core location, which suggests that our sampling location underestimates changes to the global $[NO_3^-]$ inventory.

Nevertheless, in order to prove coupling between $\delta^{13}C_{DIC}$ and $[NO_3^-]$ in intermediate water depths at different locations on glacial/interglacial timescales, a systematic downcore comparison of benthic $\delta^{13}C_{FORAM}$ and the pore density of B. spissa needs to be extended. Although the presence of B. spissa is limited to the Pacific, it occurs both on the Eastern and Western Pacific continental margin[17,53–55]. The biogeochemical model results of our study revealed no significant difference of the $\delta^{13}C_{DIC}$–$[NO_3^-]$ correlation between the LGM and the pre-industrial Holocene at intermediate water depths of the Pacific. Whereas all evidence is pinpointing that the $\delta^{13}C_{DIC}$–$[NO_3^-]$ correlation remained stable at Pacific intermediate depths on Glacial-Interglacial timescales, the validity of this correlation in other basins, on different timescales or greater water depths is not yet constrained. We therefore caution to use this correlation on a global scale before additional work has been done on testing this proxy approach in different ocean basins.

## Methods

**Stratigraphy of sediment core M77-2 52-2.** Core M77/2-52-2 was recovered during R/V Meteor cruise M77/2 in 2008[56]. The age model and $\delta^{18}O$ and $\delta^{13}C$ data of sediment core M77/2-52-2 has already been published[57] and added to the appendix. Volume defined samples were taken using cutoff syringes at 10 cm intervals. The samples were wet sieved on a 63−$\mu$m screen. The remaining >63 $\mu$m fraction of the samples was dried at 50 °C, weighed and stored for further analysis. Stable oxygen isotope ($\delta^{18}O_{FORAM}$) measurements of the cores were done with three to six individuals of benthic foraminiferal species Uvigerina peregrina. The tests of single species were crushed. Isotopic measurements were done with a Thermo Scientific MAT253 mass spectrometer equipped with an automated CARBO Kiel IV carbonate preparation device at GEOMAR, Kiel. Isotope values were reported in per mil (‰) relative to the VPDB (Vienna Pee Dee Belemnite) scale and calibrated vs. NBS 19 (National Bureau of Standards) as well as to an in-house standard (Solnhofen limestone). Long-term analytical accuracy (1-sigma) for $\delta^{18}O_{FORAM}$ and $\delta^{13}C_{FORAM}$ was better than 0.06‰ and 0.03‰ on the VPDB scale. The data for $\delta^{18}O$ and $\delta^{13}C$ is listed in the Supplementary Table 1.

For the stable nitrogen isotope ($\delta^{15}N$) measurements, 10–15 mg of freeze dried bulk sediments were analyzed using a Thermo Scientific Flash 2000 Elemental Analyzer coupled to a Thermo Scientific Delta V Advantage isotope ratio mass spectrometer (IRMS) at NIOZ, Texel. Results were expressed in standard $\delta$-notation relative to atmospheric $N_2$, and the precision as determined using laboratory standards calibrated to certified international reference standards were <0.3‰. The data for $\delta^{18}O$, $\delta^{13}C$, and $\delta^{15}N$ are listed in the Supplementary Table 1.

The already published age model is based on five $^{14}C$ AMS dating measurements performed at Beta Analytic, Inc., Florida, USA, on the planktonic foraminifera species Neogloboquadrina dutertrei[57]. Conventional radiocarbon datings were calibrated applying the marine calibration set Marine 13[58] and using the software Calib 7.0[59]. Reservoir age of 102 yrs was taken into account according to the marine database (http://calib.qub.ac.uk/marine/). Ages are expressed in thousands of years (kyr) before 1950 AD (abbreviated as cal kyr BP). The radiocarbon based chronology of the core was supplemented and tuned using Analyseries software with $\delta^{18}O$ record from a nearby core M77/2-059-1[60] and the Antarctic EPICA $\delta^{18}O$ reference stack[61].

**Quantitative $[NO_3^-]$ record using foraminiferal pore density.** Depending on the availability 7–22 Bolivina spissa specimens were picked of the >63 $\mu$m fraction from each sample of sediment core M77/2-52-2. All of the 819 specimens were mounted on aluminum stubs by the use of adhesive carbon pads. They were not sputter coated to preserve them for future geochemical analyses. Scanning electron micrographs were produced for every single individual using a Hitachi table top scanning electron microscope (TM3000 accelerating voltage of 5–15 kV and using back-scattered electrons (BSE) detector. Following the method published in ref. [17] to minimize ontogenetic effects we determined the pore density of the first (youngest) ten chambers only, relating to an area of about 50,000–60,000 $\mu m^2$. For the calculation of $[NO_3^-]_{BW}$ we used the data shown in Fig. 7d of ref. [17]. We plotted the data inversely to achieve a function in the form of Eq. 2:

$$[NO_3^-]_{BW} = a \cdot PD + b \qquad (3)$$

where $[NO_3^-]_{BW}$ is the reconstructed bottom water nitrate concentration and PD is the pore density of B. spissa. The corresponding linear fit is shown in the

Supplementary Figure 6. The resulting function (Eq. 4) has been used to quantitatively reconstruct $[NO_3^-]_{BW}$.

$$[NO_3^-]_{BW} = -3853(\pm 390) \cdot PD + 60.6 (\pm 2.2) \quad (4)$$

For the calculation of the errors for the reconstructed $[NO_3^-]_{BW}$ a complete error propagation has been done including both the uncertainty of the mean PD within the samples and the uncertainties of the calibration function. The error propagation has been applied to Eq. 3 in the form of equation Eq. 5:

$$\sigma_{[NO_3^-]_{BW}} = \sqrt{\left(\frac{\delta[NO_3^-]_{BW}}{\delta a} \cdot \sigma_a\right)^2 + \left(\frac{\delta[NO_3^-]_{BW}}{\delta PD} \cdot \sigma_{PD}\right)^2 + \left(\frac{\delta[NO_3^-]_{BW}}{\delta b} \cdot \sigma_b\right)^2}, \quad (5)$$

where $\sigma_x$ is the uncertainty (1sd) of the corresponding parameter x (in this case $[NO_3^-]_{BW}$, $a$, $b$ and PD). Considering Eq. 4 this results in Eq. 6 for the calculation of $\sigma_{[NO_3^-]_{BW}}$.

$$\sigma_{[NO_3^-]_{BW}} = \sqrt{(390 \cdot PD)^2 + (-3853 \cdot \sigma_{PD})^2 + (2.2)^2} \quad (6)$$

The standard error of the mean (SEM) for one sample was then calculated according to Eq. 7:

$$SEM_{[NO_3^-]_{BW}} = \frac{\sigma_{[NO_3^-]_{BW}}}{\sqrt{n}}, \quad (7)$$

where $n$ is the number of specimens analyzed in each sample. The results for each sample are summarized in Supplementary Table 1.

**Recent $\delta^{13}C$ on DIC and $[NO_3^-]$ in the intermediate Pacific**. All data for recent $\delta^{13}C_{DIC}$ and $[NO_3^-]$ are taken from the GLODAPv2 database[34]. The Ocean Data View (ODV) software has been used to compile the plots for Fig. 2a, b[62]. The dataset, which is shown in Fig. 2 and has been used to calculate equation 1 of the main manuscript, includes all data from 700–2000 m the recent Pacific, including parts of the Southern Ocean. Longitudinal boundaries were set to 118°E and 73°S, while latitudinal boundaries were set to 63°N and 79°S (Supplementary Figure 4A). This dataset includes 4956 measurements of both $\delta^{13}C$ on DIC and $[NO_3^-]$. Due to graphical reasons, all $\delta^{13}C$ below $-1$‰ have been cut Fig. 2c of the main manuscript. Nevertheless, all data were included into the linear fit shown in Fig. 2c and eq. 1. A complete plot of all data points can be found in Supplementary Figure 4B. Stations with low $\delta^{13}C$ mainly follow the distribution of anthropogenic $CO_2$ in the Pacific (Supplementary Note 3). The recent $[NO_3^-]$ shown in Fig. 1b has been taken from the station within the GLODAPv2 database[34] which was located closest to the location and within the same water depth of M77/2-52-2 (Station ID: 33205; Cruise: 316N19930222; Station: 356(B); 5°31'S; 85°50'W; 1278 m; $[NO_3^-] = 41.1$ μmol/l). This concentration was also used to calculate the Δ $[NO_3^-]$ for the downcore data from the pore density shown in Fig. 1c.

**Biogeochemical modeling results on $\delta^{13}C_{DIC}$-$[NO_3^-]$ coupling**. We use an improved model version of Somes et al.[7], which is based on the UVic Earth System Climate Model[63] with a version of Kiel biogeochemistry[64]. The physical ocean-atmosphere-sea ice model includes a three-dimensional ($1.8 \times 3.6°$, 19 vertical levels) general circulation model of the ocean (Modular Ocean Model 2) with parameterizations such as diffusive mixing along and across isopycnals, eddy-induced tracer advection[65], computation of tidally-induced diapycnal mixing over rough topography including sub-grid scale[66], as well as anisotropic viscosity[67] and enhanced zonal isopycnal mixing schemes in the tropics to mimic the effect of zonal equatorial undercurrents[68]. A two-dimensional, single level energy-moisture balance atmosphere and a dynamic-thermodynamic sea ice model are used, forced with prescribed monthly climatological winds[69] and ice sheets[70].

The LGM simulations were forced with boundary conditions from 21 kyr BP following the Paleo Model Intercomparison Project[71] protocols as closely as possible with our model setup. This includes lower atmospheric concentrations of the greenhouse gases carbon dioxide, nitrous oxide, and methane, changes of Earth's orbit, and the increased area and height of ice sheets[70]. The ocean grid bathymetry and total ocean volume remains unchanged relative to the pre-industrial simulation. However, effects of reduced sea level on sedimentary N loss are accounted for by calculating a new sub-grid scale bathymetry scheme assuming a constant 120 meters sea level reduction. An improved atmospheric Fe mask based on Somes et al., 2017[7] was applied by optimizing the model with $\delta^{15}N$ observations (Supplementary Figure 1). These simulations assume global $PO_4^{3-}$ and phytoplankton N:P ratios were 10% higher during the LGM.

The marine ecosystem-biogeochemical model coupled within the ocean circulation includes 2 nutrients in the inorganic ($NO_3^-$ and $PO_4^{3-}$) and organic (DON and DOP) phases, 2 phytoplankton (ordinary and $N_2$-fixing diazotrophs), zooplankton, sinking detritus, as well as dissolved $O_2$, dissolved inorganic carbon,

alkalinity, and $\Delta^{14}C$[64]. Iron limitation is calculated using monthly surface dissolved iron fields prescribed from the BLING model[72].

The $\delta^{13}C$ model is based on Schmittner and Somes[35]. The oceanic carbon cycle is governed by air-sea gas exchange of $CO_2$, which fractionates isotopes such that surface ocean DIC is ~2‰, ~8.5‰ enriched compared to the atmosphere $\delta^{13}C_{CO_2}$ = $-6.5$‰. However, fractionation during air-sea gas exchange is temperature dependent such that colder waters have higher $\delta^{13}C_{DIC}$. Uptake of DIC by phytoplankton fractionates by about $-20$‰ and depends on the pCO$_2$ of surface waters. Remineralization of the isotopically light organic carbon in the subsurface increases DIC and decreases $\delta^{13}C_{DIC}$ there. Biological production of $CaCO_3$ at the surface and dissolution at depths affects DIC and alkalinity in the model but its effect on carbon isotopes is negligible. Transient anthropogenic changes to atmospheric $\delta^{13}C_{CO_2}$ are accounted in the modern model simulation following Schmittner et al.[36].

Nitrogen isotopes are fractionated by inventory-altering ($N_2$ fixation and N loss) and internal-cycling ($NO_3^-$ uptake, excretion, DON remineralization) processes in the model[73]. $N_2$ fixation introduces isotopically light atmospheric nitrogen into the ocean ($\delta^{15}N_{Nfix} = -1$ ‰), whereas N loss fractionates strongly in the water column ($\varepsilon_{WCNl} = 20$‰) and lightly in the sediments ($\varepsilon_{SedNl} = 3.75$‰). $NO_3^-$ uptake by phytoplankton fractionates $NO_3^-$ at 6‰ in the model. Zooplankton excretion fractionates at 4‰ enriching its biomass in $^{15}N$ relative to phytoplankton. DON remineralizes with a fractionation factor of 1.5‰ to reproduce upper ocean $\delta^{15}N$-DON observations mainly occurring within the range of 3–6‰[74]. For a detailed discussion about offsets between measurements and modeling of the modern $\delta^{13}C_{DIC}$-$[NO_3^-]$-coupling see Supplementary Note 4 and Supplementary Figure 7.

**Regional offsets between Holocene and LGM $NO_3^-$ inventories**. A compilation of deglacial $\delta^{13}C_{FORAM}$ change measured downcore on tests of Cibicidoides spp. has been published in ref. [37]. We extracted all records from this compilation available from 700–2000 m using the same latitude/longitude window mentioned above (see also Supplementary Figure 4A). Relative $[NO_3^-]$ changes were calculated after equation 1 using the offset of mean $\delta^{13}C_{FORAM}$ measured on Cibicidoides spp. between LGM (19–23 kyrs BP) and Late Holocene (0–6 kyrs BP). Four downcore datasets from two additional references[75,76] were added which were not included in ref. [37]. The results are compiled in Supplementary Table 3. For the comparison of mean deglacial changes in the Pacific $NO_3^-$ inventories which is shown in Fig. 1c of the main manuscript, only the 14 stations located in the Pacific and measured on Cibicidoides spp. were used. These stations are clearly marked in Supplementary Table 3. The mean Pacific $[NO_3^-]$ was 3.0 ($\pm 0.5$ 1 SEM; N = 14) μmol/kg higher during the LGM. The mean offset between LGM and Holocene is slightly lower ($2.3 \pm 0.5$ μmol/kg (1 SEM); N = 23) if also the sampling locations outside the Pacific are included. For a regional comparison of deglacial $NO_3^-$ changes, including all stations see Supplementary Figure 3. The general trend of all stations again indicates a higher $NO_3^-$ inventory during the LGM. Only individual stations in the Sea of Japan, the East China Sea and on the Southern Australian Continental Margin indicate no changes between LGM and Late Holocene. One station at the Southern Australian Continental Margin even indicated a depletion of $NO_3^-$ during the LGM compared to the late Holocene. The highest deglacial $[NO_3^-]$ changes are located a station south of New Zealand and in the Sea of Okhotsk, which is probably related to the high latitudes of these sediment cores. The Sea of Okhotsk was still covered by ice during the LGM.

**Data availability**. All data which support the findings of this study are either available online or within the Supplementary material. The foraminiferal pore density, reconstructed $[NO_3^-]_{BW}$, $\delta^{18}O$, $\delta^{13}C$ and $\delta^{15}N$ for sediment record M77/2-52-2 is available in Supplementary Table 1. The data for $\delta^{13}C_{FORAM}$ and the reconstructed deglacial $[NO_3^-]$ offsets for all records from the intermediate Pacific is available in Supplementary Table 3. The model code and output for the 3D Biogeochemical modeling on deglacial $\delta^{13}C_{DIC}$-$[NO_3^-]$ coupling are available on the GEOMAR Thredds Server (https://thredds.geomar.de).

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

## Acknowledgements
The scientific party on R/V *Meteor* cruise M77 is acknowledged for their general support and help with the coring operation and sampling. Tal Dagan supported this manuscript with feedback about the statistical evaluation of this study. We thank Ralph Schneider for the provision of the sediment core material used for this study and Ralf Schiebel for constructive feedback on an early draft of this manuscript. Additionally, we would like to thank Joachim Oesert for support at the electron microscope and Andreas Schmittner for developing and sharing the $^{13}$C model code. Funding was provided by the Deutsche Forschungsgemeinschaft (DFG) through the SFB 754 "Climate–Biogeochemistry Interactions in the Tropical Ocean" and the European Research Council (ERC) under the European Union's Seventh Framework Program (FP7/2007-2013) ERC grant agreement [339206]. C.J.S. received support for his contribution from the PalMod project.

## Author contributions
N.G. analyzed and interpreted the foraminiferal pore density data, did the database investigation, main interpretation of $\delta^{13}$C records and core-writing of the manuscript. Z. E. provided the age model and stratigraphy and performed the $\delta^{15}$N analyses for core M77/2-55-2. K.W. provided the continuously modeled, global reactive N-inventory and contributed to the interpretation of the $\delta^{13}$C records. C.J.S. did the 3D Biogeochemical modeling on deglacial $\delta^{13}$C$_{DIC}$-[NO$_3^-$] coupling. V.L. contributed to the initial interpretation of the nitrate record. J.S. provided additional information about available $\delta^{13}$C records. S.G. provided access to the electron microscope for the determination of foraminiferal pore densities. A.E. contributed to the discussion of data and interpretation.
