## [Peer Review File(PDF 1999 kb) · Nature Communications]

Editorial Note: Parts of this peer review file have been redacted as indicated to remove third-party material where no permission to publish could be obtained.

Reviewers' comments:

Reviewer #1 (Remarks to the Author):

Review for Glock et al., Coupling of oceanic carbon and nitrogen: A window to spatially resolved quantitative reconstruction of nitrate inventories. Glock et al. uses benthic foraminifera pore density and the carbon isotopes to infer larger nitrate inventories in the last glacial maximum. This is an important and interesting biogeochemical question, but the paper fails to present a strong data based argument. Based on my following concerns, I would recommend rejection.

The most critical mistake this paper made is the use of modern and paleo correlation between carbon isotope and nitrate concentration to infer past changes in nitrate inventory. The correlation between carbon isotope and nitrate concentration is expected to be similar in the present and past because this is essentially driven by redfield-remineralization. But this does not constrain the intercept of the correlation where nitrate concentration strongly depends on the preformed nitrate in the polar ocean, but carbon isotope is strongly influenced by the carbon and carbon isotope distribution between the ocean and the land/atmosphere. The authors interpret the changes in nitrate concentration in the Pacific as a result of the input/output processes in N inventories (Line 168 and 169). This is wrong.

So, my interpretation of this record solely relies on the use of benthic foraminifera pore density. I am not familiar with this proxy. While this paper refers to reference 15, I suggest a simple explanation of this proxy is needed in the main text. Why nitrate concentration not oxygen concentration determines the pore density? To which extent has this proxy been quantitatively calibrated? Is the past changes in oxygen and nitrate concentration within the range of calibration?

Assuming the interpretation is correct that greater pore density results from higher benthic nitrate concentration, this paper needs much more to interpret this as higher nitrate inventories in the glacials. At any given location, the nitrate concentration change could result from (1) local processes, such as direct influence from OMZ especially given the core's location; (2) preformed nitrate concentration which varies as a function of polar ocean nutrient consumption; (3) remineralized nitrate concentration which is controlled by the rate of remineralization occurring along the circulation path as well as the age of the intermediate water. Out of these processes, (1) is strongly influenced by oxygen content, local productivity and ocean circulation. While the authors argue that the different timing between their $\delta^{15}\text{N}$ change and pore density change suggest the benthic nitrate concentration is not directly related to OMZ influence, this is not a convincing argument. $\delta^{15}\text{N}$ changes speaks to the shallow OMZ where denitrification fractionation on nitrate can be reflected by surface sinking particles, so it may indeed be the case that the shallow intermediate OMZ was smaller during the last ice age. However, many evidence based on redox sensitive proxies suggests that the deep intermediate and bottom waters of the Pacific ocean was more anoxic during the last ice age, arguing that potentially there could be a shift of water column denitrification to deeper depth. But this may not be well captured by $\delta^{15}\text{N}$ changes, as the deeper denitrification signal is less well communicated with the surface ocean. At the minimum, I think this paper should look for changes of redox sensitive proxies at or near this core location. (2) the preformed nitrate concentration is likely to be lower during the last ice age, so it cannot help to explain the greater nitrate concentration. However, I still need to point out that the lower ice age preformed nitrate is not because of "enhanced burial of preformed nitrate in AABW through export productivity (Line 95-96)". Higher ice age export productivity is only observed in the subantarctic. The Antarctic export productivity is in fact lower during the ice age possibly due to stronger stratification and less wind driven upwelling. (3) the remineralized nitrate concentration is expected to be greater in the ice age, given the coupled oceanic carbon and nitrogen cycle, if the ocean is responsible for carbon burial, then it requires more remineralized nitrate. But how much of this increase is due to whole ocean nitrate inventory change is difficult to answer. At this specific location, I would like to at least look at the ventilation age of the bottom

water. In summary, I do not think this paper has well addressed the processes that could contribute to changes in bottom water nitrate concentration, so their conclusion for changes in global nitrate inventory is too speculative to be the main findings of this study.

Reviewer #3 (Remarks to the Author):

This manuscript addresses a highly relevant topic, the reconstruction of nitrate concentrations in ancient ocean basins. This is of great importance, since nitrate plays a key role in the global nitrogen cycle and heavily influences the global carbon cycle. As such the nitrogen cycle is present in global climate models. To date, there is no faithful proxy to reconstruct past nitrate concentrations, needed for the tuning of the models.

The method presented here to estimate past nitrate concentrations, by using benthic foraminiferal pore density, is very innovative, elegant, and convincing, at least at a regional scale. It is absolutely the first time that a method capable to reconstruct nitrate concentration in oceanic bottom waters is presented. As such, the paper is of large interest for the whole paleoceanographic community, and the ideas will certainly be taken up by others. Consequently, in my opinion it should absolutely be published. The paper is fairly mature; in my opinion only a moderate revision is necessary, in which the following aspects could/should be considered:

- 1) The authors have a tendency to overstretch the consequences of their results. They tend to forget that their results, although being extremely valuable, pertain only to intermediate water masses in a small part of the Pacific, with a well developed OMZ. They don't have any proof that that foraminiferal pore density will be perfectly correlated with nitrate concentration in other settings, or in other ocean basins. The same holds true for the correlation between nitrate concentration and $\delta^{13}\text{C}$. I think that they should at several places be slightly more conservative in their writing.
- 2) Related to this, the authors should try to be as precise as possible in their use of terminology and of English. For instance, $\delta^{13}\text{C}$ (an isotope ratio) can't be "enriched". When they talk about the $\delta^{13}\text{C}$ of bottom water DIC (not foram $\delta^{13}\text{C}$), they should consistently write $\delta^{13}\text{C}_{\text{DIC}}$. More suggestions for a more precise usage of terminology are given in the annotated manuscript.
- 3) I am surprised, when the authors discuss the very consistent correlation between $\delta^{13}\text{C}_{\text{DIC}}$ and nitrate concentration in intermediate water masses, the authors don't evoke a possible control of ventilation (speed of renewal of bottom waters). In my opinion, ventilation could very well be the "the background driver controlling both parameters". If true, this would mean that the correlation observed here is valid for intermediate waters of the whole Pacific, but the relation could be less robust in other basins, and the exact calibration will almost certainly be different. I invite the authors to reflect on this.
- 4) In fact, the authors tentatively present $\delta^{13}\text{C}$ of epibenthic foraminifera as a second proxy (next to foraminiferal pore density) to reconstruct ancient nitrate concentrations in intermediate waters in the Pacific. The two proxies are totally independent. Although Fig. 1B already shows that the two downcore records (reconstructed nitrate pore density and benthic $\delta^{13}\text{C}$) are very similar, it would be interesting to compare the two nitrate proxies directly, by adding in Fig. 1C a curve with reconstructed nitrate using equation 1.

Reviewer #4 (Remarks to the Author):

Review of "Coupling of oceanic carbon and nitrogen: A window to spatially resolved quantitative

reconstruction of nitrate inventories" by Glock et al.

Summary

The authors use the pore-density (number of pores per unit area) in benthic forams as a proxy for bottom water NO_3^- concentration. From this, they recreate a record of bottom water NO_3^- concentration from the modern to LGM using a sediment core from the eastern Pacific (Peruvian continental margin). They then compare this record to those of $\delta^{13}\text{C}$, $\delta^{18}\text{O}$ of foram tests, with model predictions of $[\text{NO}_3^-]$ and with $\delta^{15}\text{N}$ of bulk organic material over the same sediment core. Among their findings – they highlight a close correspondence in the relationships between $\delta^{13}\text{CDIC}$ and $[\text{NO}_3^-]$ at intermediate water depths in the modern ocean and that between downcore $\delta^{13}\text{C}_{\text{foram}}$ and $[\text{NO}_3^-]$ derived from foram pore-density measurements. Since these two relationships appear so similar – the authors propose that downcore measurements of $\delta^{13}\text{C}_{\text{foram}}$ can be used to reconstruct bottom water NO_3^- concentrations – and describe data that appear to corroborate this conclusion. Using this approach – the authors estimate that bottom water $[\text{NO}_3^-]$ in this region was $\sim 3\mu\text{M}$ higher during the LGM – which is also consistent with some modeling studies. The bottom water $[\text{NO}_3^-]$ record presented has a much higher temporal resolution than other approaches – offering unique and interesting insights into mechanisms of C and N cycling and their coupling. Two notable "side-aspects" are also included. Given the increased temporal resolution – the authors point out a brief period in which the record of bottom water $[\text{NO}_3^-]$ appears to diverge from modeled NO_3^- and offer water mass mixing effects ('local effects') as the primary mechanism for this. Second, the authors point out that the tight coupling between NO_3^- and DIC dynamics over the LGM to present interval – offers a potentially new opportunity for constraining the relative contribution of anthropogenic CO_2 .

Overall, I was very impressed with the apparent fidelity of the record from pore-density measurements and with the authors' examination of its historical record in the context of glacial/interglacial time intervals for these depths in the Pacific. This record provides a new temporal resolution for estimation of bottom water NO_3^- at intermediate depths and could be applied to deepen our understanding of carbon/nitrogen coupling and glacial/interglacial dynamics in a more spatially and temporally resolved manner. I would recommend publication after addressing my (mostly minor) comments below.

Major comments

1. The manuscript is very well written in general – but suffers from some English usage errors. I attempted to correct these or make suggestions in some places, but feel that the manuscript would benefit from a more thorough review for language clarity.
2. The paper rests very heavily on work previously published by these authors (e.g., Glock et al., 2011). While I have no reason to doubt the veracity of these previous studies – I think that the average reader would benefit from a slightly more detailed explanation for the mechanism underlying pore-density response to nitrate – presented in the main text. While this is addressed to some degree in the Supplemental Material (Sup 4) – I feel that even just 2-4 sentences here that explain the underlying assumptions, mechanisms and limitations, would be quite useful. Also, to this point, is it possible that this relationship is more directly linked to O_2 stress (and thus, indirectly to NO_3^-) in the modern ocean? Where O_2 and NO_3^- are decoupled would this interfere in the fidelity of this proxy? For example, the relationship between $\delta^{13}\text{CDIC}$ and NO_3^- holds up for intermediate depths (this is well demonstrated by the presented data) – but could changes in hydrography decouple this relationship to some degree? I think that this is what is referred to in lines 106-109 and discussed in the Supplemental Material (Sup 1)? In this respect – it seems that great care is required for recognizing such shortcomings ('local effects'), and this aspect should be emphasized.
3. The section discussing the coupling of $\delta^{13}\text{CDIC}$ and $[\text{NO}_3^-]$ is interesting - in particular, the possibility of related feedbacks controlling the relative proportions of N-fixation and denitrification

– and implications for the global N budget. However, it was not made clear how these feedbacks (and the throttling up of benthic denitrification) would operate to modify $\delta^{15}\text{N}$, when benthic denitrification is not considered to impact the $\delta^{15}\text{N}$ of seawater NO_3^- . Perhaps I am misunderstanding something in your argument – but in either case I think there could be some attention given to improving the clarity of these arguments.

4. What is the main purpose of introducing the data from the *Cibicides* records at the very end? This exercise seems a little out of place in the presentation. Do you mean to demonstrate that the approach works for $\delta^{13}\text{C}$ records from carbonates in other species? Not clear.

5. It doesn't seem to me that the mid-Holocene minimum in bottom water $[\text{NO}_3^-]$ is significantly different than the other points from within the mid/late Holocene. In other words – I'm not sure it needs to be singled out as a 'feature' (L107-109; L455-458).

6. Several important points are described by the authors in the Supplemental Material – including the influence of microhabitat preferences on foram $\delta^{13}\text{C}$, and the nature of pore-density in benthic forams, etc. While I understand the restrictions of space for publication in Nature Communications, I think it would be useful to at least briefly describe some of these points in a bit more detail – especially the topics covered in Sup 3 and Sup 4. If you are cut short on space, I think that making readers more comfortable with your assumptions of the approach might be more important than the approach for anthropogenic CO_2 (L161-165).

Minor comments

I would encourage the authors to use appropriate subscripts on their isotope notation for clarity. For example, the $\delta^{13}\text{C}$ values of DIC (whether measured from water or inferred from foram carbonate) would benefit throughout by a subscript (e.g., $\delta^{13}\text{C}_{\text{DIC}}$) to aid in distinguishing between $\delta^{13}\text{C}$ of foram carbonate tests and $\delta^{13}\text{C}$ of DIC.

Similarly, I feel that the N isotopes should be explicitly denoted as referring to the bulk sediment organic matter (e.g., $\delta^{15}\text{N}_{\text{sed}}$, $\delta^{15}\text{N}_{\text{bulk}}$, or $\delta^{15}\text{N}_{\text{NorgM}}$, etc.) – to make it clear that these are not somehow referring to measurements of foram bound N or NO_3^- .

Figure 1 – it would be very helpful if light lines could be added (through panels A through D) that guide the eye for the discussed time periods.

Would equation 1 be better presented flipped over and solving for past $[\text{NO}_3^-]$ here?

L30: 'is considered to the ongoing ocean deoxygenation' – awkward – consider "is considered to be a large contributor to ongoing ocean deoxygenation."

L39: I would suggest the use of "oxygen deficient zones (ODZs)" – since oxygen minimum zones (OMZs), which occur across most water columns throughout the ocean, simply reflect elevated rates of respiration – yet may not coincide with zones of N loss via denitrification. The main point is that ODZs are regions of N loss. I realize that OMZs has been widely used to indicate the three major zones of the global ocean where N loss occurs – but feel that a more precise use of the terminology should be invoked.

L89-90: This sentence is not clear to me.

L99: I find the phrase 'burial of preformed nutrients' somewhat peculiar. "Burial of organic matter?"

L107: I think you should rephrase "This offset and the minimum of $[\text{NO}_3^-]$ in our sediment record that appears during the middle Holocene from..." Otherwise this sounds like both phenomena occur

at the same time – which I believe is not the intended meaning.

L116: superfluous “in”

L117: Please indicate which specific time interval you are referring to here.

L118: Either use “While” or “whereas” – but not both in the same sentence. Omit one or the other.

L120: indicates

L128: Is there a straightforward reason that $\delta^{13}\text{C}$ measurements were NOT made on *B. spissa*?

L138: established

L157: foraminifera

L169: “...do not influence $\delta^{13}\text{C}$ in the same direction.”

L171: “The decreased area of continental shelves during the LGM in comparison...”

L173: I think this ‘negative’ feedback should actually refer to a ‘positive’ feedback (?). I believe you are saying that if N-fixation goes up – so does denitrification? This would be a positive feedback. If this is not the case – then please clarify.

L191: “Nevertheless, in order to prove coupling...”

Reference #6 is repeated as Ref #18.

Figure 1: Panel A – the $\delta^{18}\text{O}$ data is never discussed.

Figure 1: Panel B – the green squares are very difficult to see.

Figure 1: Panel C is misaligned with the other panels.

Figure 1: Panel D – should included error bars and refer to $\delta^{15}\text{N}$ as “ $\delta^{15}\text{N}$ of sediment organic matter.”

Methods:

Equation 4 – what do the small ‘deltas’ refer to?

L378: I think you mean Eq 6 here?

L389: ...were included into the...

L445: ...surface water into intermediate...

L488: ...‘why’ there is no...

Index:

Answers to the comments of Reviewer #1	Page 1
Answers to the comments of Reviewer #3	Page 11
Answers to the comments of Reviewer #4	Page 14
Reference list for the Response Letter	Page 20

Answers to the comments of Reviewer #1

We thank the reviewer for the effort reviewing our manuscript. The points of revision were very critical but also very constructive. Below you can find a point-by-point response to the review. We addressed all the issues brought up by Reviewer #1 and in our opinion the revised version of manuscript has substantially improved, due to the constructive feedback of the reviewer. The manuscript was extended to discuss alternative scenarios, like local ODZ fluctuations and deglacial changes in the ventilation of the Pacific. We added some new results, modeling the distribution of $\delta^{13}\text{C}$ and $[\text{NO}_3^-]$ for the modern ocean, the pre-industrial Holocene and the LGM using a coupled 3D ocean circulation-biogeochemical-isotope model. The model system used here is an improved version of Somes et al. (2017) by including the carbon isotope cycling following Schmittner and Somes (2016). The complete reference list which has been used for this review can be found at the end of this letter after the response to all reviewers.

Reviewers' comments:

R#1: Reviewer #1 (Remarks to the Author):

Review for Glock et al., Coupling of oceanic carbon and nitrogen: A window to spatially resolved quantitative reconstruction of nitrate inventories. Glock et al. uses benthic foraminifera pore density and the carbon isotopes to infer larger nitrate inventories in the last glacial maximum. This is an important and interesting biogeochemical question, but the paper fails to present a strong data based argument. Based on my following concerns, I would recommend rejection.

The most critical mistake this paper made is the use of modern and paleo correlation between carbon isotope and nitrate concentration to infer past changes in nitrate inventory. The correlation between carbon isotope and nitrate concentration is expected to be similar in the present and past because this is essentially driven by redfield-rem mineralization. But this does not constrain the intercept of the correlation where nitrate concentration strongly depends on the preformed nitrate in the polar ocean, but carbon isotope is strongly influenced by the carbon and carbon isotope distribution between the ocean and the land/atmosphere.

Reply: The fundamental parameters controlling the $[\text{NO}_3^-]$ and $\delta^{13}\text{C}_{\text{DIC}}$ were already briefly reflected in the original manuscript. The reviewer is certainly right that we discussed them less than necessary for a comprehensive understanding. Below, we discuss in detail the three points which, according to the reviewer, might control the intercept of the correlation between $\delta^{13}\text{C}_{\text{DIC}}$ and $[\text{NO}_3^-]$. Furthermore, our new biogeochemical modelling results show, that the correlation, neither intercept nor slope,

did not significantly change between the LGM, the pre-industrial Holocene and the modern ocean.

R#1: "...but carbon isotope is strongly influenced by the carbon and carbon isotope distribution between the ocean and the ... atmosphere."

Reply: It is true that the intercept of the correlation might be also controlled by the carbon and carbon isotope distribution between the ocean and the land/atmosphere which are independent from $[NO_3^-]$ concentrations. Nevertheless, recent modelling studies showed that lower CO_2 concentration in the glacial atmosphere compared to the Holocene and thus lower carbon exchange between ocean and atmosphere at glacial times cannot explain the lighter stable carbon isotope inventory during the LGM (Galbraith et al., 2015). Despite that, the authors took caution to use $\delta^{13}C$ as a nutrient proxy. The pCO_2 effect might mask the influence of the biological pump on $\delta^{13}C_{DIC}$ if the pCO_2 gradient is very strong at times of high atmospheric pCO_2 such as during the early Cenozoic.

To address this issue, we added the following part into the revised manuscript:

"A factor controlling the mean $\delta^{13}C_{DIC}$ in seawater is the exchange of atmospheric CO_2 with the ocean surface. A change in atmospheric pCO_2 would also mediate disequilibrium in the surface ocean. However, recent studies showed that this " pCO_2 effect" would cause a maximum $\delta^{13}C_{DIC}$ offset in subsurface waters of the Southern Ocean of approximately 0.2‰ (Galbraith et al., 2015). This deglacial offset is even smaller in other parts of the oceans and close to zero at our sampling location and thus cannot explain the changes of $\delta^{13}C_{Foram}$ in our downcore record. Despite this low deglacial offset in $\delta^{13}C_{DIC}$ by the pCO_2 effect the authors of named study caution to interpret $\delta^{13}C_{Foram}$ as a nutrient proxy. The pCO_2 effect might mask the influence of the biological pump on $\delta^{13}C_{DIC}$ if the pCO_2 gradient is very strong at times of high atmospheric pCO_2 such as during the early Cenozoic."

Furthermore, we added the following part to the revised manuscript to point out that the situation might be different in other environments or on different timescales:

"Our record and the comparison to the other deglacial $\delta^{13}C_{FORAM}$ records is considered as evidence that coupling of $[NO_3^-]$ and $\delta^{13}C_{DIC}$ was mainly controlled by the biological carbon pump at our sampling location and intermediate depths of the Eastern Equatorial Pacific, and possibly in other Pacific regions and intermediate depths. The situation might be different in the Atlantic Ocean, at greater depths or further back in Earth's history. Independent calibrations are thus substantial to extend the application of $\delta^{13}C_{Foram}$ as a quantitative $[NO_3^-]$ proxy. Furthermore, it is unlikely that this proxy would work at shallow depths where the pCO_2 effect might predominate the effect of the biological carbon pump."

R#1: "...but carbon isotope is strongly influenced by the carbon and carbon isotope distribution between the ocean and the land/... "

Reply: We understand the reviewer's opinion is that the intercept of the $[NO_3^-]$ - $\delta^{13}C_{DIC}$ correlation might be controlled by the transport of terrestrial carbon to the ocean. Indeed, the glacial $\delta^{13}C_{DIC}$ depletion has been widely ascribed to a glacial loss of terrestrial biomass (Shackleton, 1977; Köhler and Fischer, 2004). However, recent model studies can reproduce almost the entire glacial shift to depleted $\delta^{13}C_{DIC}$ values recorded in benthic foraminifera (0.34 ± 0.19 ‰; Peterson et al., 2014) without invoking any net changes in terrestrial biomass (Wallmann et al, 2016). This outcome is consistent with our analytical results, and in agreement with the results of a new

modelling study suggesting that the rise in carbon buried in permafrost and under ice largely compensated for the decline in peat, soil, and biomass carbon over the LGM (Brovkin and Ganopolski, 2015).

R#1: “But this does not constrain the intercept of the correlation where nitrate concentration strongly depends on the preformed nitrate in the polar ocean...”

Reply: We disagree with this point, because the concentration of preformed NO_3^- is not independent from $\delta^{13}\text{C}_{\text{DIC}}$. The term “preformed nutrients” is defined as the amount of nutrients (e.g. nitrate or phosphate) which is not utilized in the thermocline (Broecker, 1974; Broecker and Peng, 1982; Broecker et al., 1985). This does not make them independent from the biological pump. If less nutrients are utilized, the biological pump is reduced which results in higher remaining nutrient concentrations accompanied by $^{12}\text{C}_{\text{DIC}}$ enriched waters since less $^{12}\text{C}_{\text{DIC}}$ is removed from the surface waters by the export of particulate organic matter into deeper waters, *i.e.* the biological pump. Thus, any change in preformed nutrients would just shift the data points parallel to the $[\text{NO}_3^-]$ - $\delta^{13}\text{C}_{\text{DIC}}$ correlation line.

As already mentioned above, our new biogeochemical modeling results support our previous interpretations of analytical data. The model reveals that neither intercept nor slope of the $\delta^{13}\text{C}_{\text{DIC}}$ - $[\text{NO}_3^-]$ -correlation changed significantly between the LGM, the pre-industrial Holocene and the modern ocean. The following part has therefore been added to the discussion of the revised manuscript:

*“The distribution of $\delta^{13}\text{C}$ and $[\text{NO}_3^-]$ has been modeled for the modern ocean, the pre-industrial Holocene and the LGM (Fig. 3) using a coupled 3D ocean circulation-biogeochemical-isotope model. The model system used here is an improved version of Somes et al. (2017) by including the carbon isotope cycling following Schmittner and Somes (2016) and optimizing LGM iron deposition patterns to better reproduce $\delta^{15}\text{N}_{\text{bulk}}$ observations (see Supl. Fig 7). The modeling results indicate no significant difference in the relationship of the $\delta^{13}\text{C}_{\text{DIC}}$ - $[\text{NO}_3^-]$ -correlation in the deep intermediate Pacific at our core location (*i.e.* $[\text{NO}_3^-]_{\text{BW}}$ μM ; Supp. Fig 6) during the different climatic time intervals. This supports our comparison of the M77/2-52-2 sediment record to the modern $\delta^{13}\text{C}_{\text{DIC}}$ - $[\text{NO}_3^-]$ distribution.*

Additionally, the model results support that $[\text{NO}_3^-]$ at our sampling location records changes in the global budget (predicted at our location: $\Delta\text{NO}_3^- = 3.0 \mu\text{M}$), but also is affected by iron fertilization at high latitudes. Iron fertilization decreases preformed nutrients in SAMW and shallow AAIW, where our core location exists, of the Pacific due to the transfer of more remineralized nutrients to the deep Pacific. This process is observationally constrained in the 3D model by direct comparison to $\delta^{15}\text{N}_{\text{bulk}}$ across the Southern Ocean (Supp. Fig 7), which records changes to surface $[\text{NO}_3^-]$ utilization in response to dust deposition (Somes et al., 2017). Sensitivity simulations associated with Southern Ocean iron fertilization uncertainties cause $[\text{NO}_3^-]$ changes at our location of $\pm 0.7 \mu\text{M}$ on top of the direct impact on global $[\text{NO}_3^-]$ (see Sup. Table 2). The increase to global $[\text{NO}_3^-]$ in the model was $1.5 \mu\text{M}$ larger than bottom water $[\text{NO}_3^-]$ change at our core location, which suggests that our sampling location underestimates changes to the global $[\text{NO}_3^-]$ inventory.”

For further details of the model see the methods section and supplementary (Sup. 4) of our revised manuscript. The main modeling results are shown in figure 3 of the revised

manuscript. The model also covers This includes lower atmospheric concentrations of the greenhouse gases carbon dioxide, nitrous oxide, and methane, changes of Earth's orbit, and the increased area and height of ice sheets during the LGM (Peltier, 2004).

R#1: The authors interpret the changes in nitrate concentration in the Pacific as a result of the input/output processes in N inventories (Line 168 and 169). This is wrong.

Reply: We concede that we probably did not discuss all local factors in detail which might have influenced the nitrate concentrations at our sampling location over the last deglaciation. In the original manuscript, we had a chapter in the Supplement, discussing local nitrate variability (sup. 1). In the revised manuscript we discuss alternative scenarios which might have led to increased nitrate concentrations at our sampling location (e.g. deep Pacific ventilation and local ODZ variability). We also added C_{org} accumulation rates from our record as an additional proxy for paleoproductivity. Nevertheless, all parallel changes (i.e. lower $\delta^{13}C_{FORAM}$, higher nitrate, more oxygen and lower primary productivity) can only be explained if nitrate concentrations are mainly influenced by changes in the global inventories. Decreased ventilation might explain the lower $\delta^{13}C_{FORAM}$ and the higher nitrate but is contradicting the higher oxygenation at these water depths during the LGM. A less pronounced ODZ during the LGM could explain the higher nitrate and higher oxygen levels but not the lower $\delta^{13}C_{FORAM}$ of our record. Besides, nutrient utilisation was relatively high and did not substantially change during the last deglaciation above our sampling location (Doering et al., 2016). The good quantitative agreement of our reconstructed nitrate concentrations to the predictions of inventory changes from global biogeochemical models is intriguing, though. There was one factor we missed in our interpretation so far. As mentioned above, we added new modelling results to our study. According to these results nitrate concentrations (and $\delta^{13}C_{FORAM}$) at intermediate water depths of the Eastern Equatorial Pacific were not only influenced by the global inventories along the last deglaciation but also by changes in iron fertilization at high latitudes. This resulted in a decrease of nutrients in shallower water depths of the Pacific due to the transport of more nutrient depleted AAIW to lower latitudes but also in an enrichment of nutrients in the intermediate to deep Pacific due to increased stratification. This is now discussed in the revised manuscript (see above).

R#1: So, my interpretation of this record solely relies on the use of benthic foraminifera pore density. I am not familiar with this proxy. While this paper refers to reference 15, I suggest a simple explanation of this proxy is needed in the main text. Why nitrate concentration not oxygen concentration determines the pore density? To which extent has this proxy been quantitatively calibrated? Is the past changes in oxygen and nitrate concentration within the range of calibration?

Reply: In the original manuscript we already provided a more detailed explanation of this proxy in the Supplement (S4). Since all the reviewers agreed that an explanation of this proxy is also needed in the main text, we moved this part from the Appendix to the main text. Furthermore and since the reviewer asked to which extent the proxy has been quantitatively calibrated, we added a sentence about the amount of available proxy calibration data. The proxy has been calibrated to bottom water nitrate concentrations on 8 different locations from the Peruvian continental margin with 232 measurements of pore density on individual specimens of *Bolivina spissa*.

The following part has been added to the main manuscript after the introduction:

“The functionality of pores in benthic Foraminifera

A comprehensive review about the functionality of pores in benthic foraminifera can be found in Glock et al., 2012. The functionality of pores in Foraminifera reaches from purely ornamental (Angell, 1967) to gas exchange for the uptake of electron acceptors and the release of metabolic waste products like CO₂ (Leutenegger and Hansen, 1979) until the uptake of dissolved organic material (Berthold, 1976). Foraminifera from oxygen depleted environments typically show an increased porosity (Kaiho, 1994) and often a clustering of mitochondria behind the pores (Leutenegger and Hansen, 1979, Bernhard and Bowser, 2010). Several recent studies describe the influence of oxygen availability on foraminiferal pore characteristics (Kuhnt et al, 2013; Kuhnt et al., 2014, Petersen et al., 2016). While some species adapt their porosity by changing the size of their pores (Petersen et al, 2016), other species are adapting the numbers of pores (pore-density) in their tests (Glock et al., 2011, Kuhnt et al, 2013; Kuhnt et al., 2014).

*Benthic Foraminifera from oxygen depleted environments have recently been shown to use NO₃⁻ as electron acceptor (Risgaard-Petersen et al., 2006, Pina-Ochoa et al., 2010). At least one species, *B. spissa*, from the Peruvian ODZ, adapts its pore density to the availability of NO₃⁻ in its habitat (Glock et al., 2011). A comparison of 232 measurements of the pore density in *B. spissa* to the bottom water nitrate concentrations ([NO₃⁻]_{BW}) from 8 different sampling locations at the Peruvian continental margin revealed a significant linear relationship between both parameters. Another species from the Peruvian ODZ, *Bolivina seminuda*, has been shown to have a high affinity to NO₃⁻ availability (Cardich et al., 2015). The tests of *B. seminuda* are highly porous (Glock et al., 2011). Every species of the genus *Bolivina* which has been analysed so far, including *B. seminuda*, has the ability to denitrify (Pina-Ochoa et al., 2010, Bernhard et al., 2012), which implies that denitrification is a common strategy of Bolivinidae for survival under oxygen depleted conditions. making species from this genus particular candidates for paleo NO₃⁻ reconstruction by analyses of their pore characteristics as an empirical proxy.”*

Furthermore, recent studies at the Peruvian ODZ showed that *Bolivina spissa* indeed denitrifies. It has a very low ratio of oxygen respiration/denitrification compared to other foraminiferal species living at the upper boundary of the Peruvian OMZ, where oxygen intrusions are much more common than in the habitat of *B. spissa* at the lower OMZ boundary (own data). This indicates that *B. spissa* indeed is much more dependent on nitrate instead of oxygen as an electron acceptor.

We agree with the reviewer's comment about the range of calibration. A part of the reconstructed [NO₃⁻]_{BW}, is outside the calibration range. In particular values from the LGM and the deglaciation were extrapolated by 3-4.5 μmol/l from the range of modern calibration. A complete error propagation, including the error of the calibration function, has been calculated for each datapoint (see eq. 2-6 of the original manuscript). Thus, the accuracy of the extrapolated datapoints is captured by the confidence bands, although reflected by wider error ranges for the extrapolated datapoints. The correlation between the pore density and [NO₃⁻]_{BW} in the original manuscript was significant over the total concentration range from 18 to 41 μmol/l (a range of about 23 μmol/l; Glock et al., 2011). It is therefore unlikely that the correlation is completely different within a narrow extrapolation range of 3-4.5 μmol/l. Furthermore, for our paleo-record we focused on a linear correlation starting from 34 μmol/l (Glock et al., 2011; Fig. 7d), which is closest to the range of the pore densities found within the studied core. The very low pore densities, and thus the high corresponding [NO₃⁻]_{BW} during the LGM are unprecedented at modern continental margin environments in the Pacific, where

Bolivina spissa lives today. The extrapolation of the modern calibration by a few $\mu\text{mol/l}$ for the interpretation of the data is, in our opinion, a valid and accurate approach. Its uncertainties are covered by the shown errors arising from the confidence bands of the proxy calibration. Furthermore, the above interpretation is corroborated by the $[\text{NO}_3^-]$ of different global biogeochemical models, as discussed in the main text, and the interpretation of $\delta^{13}\text{C}_{\text{Foram}}$ as an additional independent proxy. Finally, the fact that such low pore-densities as during the LGM have not been recorded at the modern Peruvian Margin, again indicates the strong biogeochemical difference in this environment during glacial times. The correlation between pore density of *B. spissa* and bottom-water oxygen concentration is exponential (decay), reaching a plateau at a pore density of ~ 0.005 pores μm^{-2} where the pore density becomes independent from oxygen concentrations. The fact, that the pore density found in LGM sediments is even lower further supports its interpretation as $[\text{NO}_3^-]_{\text{BW}}$ proxy. For the matter of simplification for the review process we add here the complete calibration graphs of nitrate vs pore density from Glock et al., 2011 (see Fig R1):

[Redacted]

**Bottom-water nitrate
concentration ($\mu\text{mol/l}$)**

Fig. R1: From Glock et al., 2011 (Figure 7c&d). Pore density in *B. spissa* vs. $\text{NO}_3\text{-BW}$. N represents the number of individually analysed specimens of *B. spissa* for the pore density measurements.

R#1: Assuming the interpretation is correct that greater pore density results from higher benthic nitrate concentration, this paper needs much more to interpret this as higher nitrate inventories in the glacials. At any given location, the nitrate concentration

change could result from (1) local processes, such as direct influence from ODZ especially given the core's location;

Reply: Local ODZ variability and other processes had briefly been discussed in the original manuscript (sup.1). This discussion is now extended in the revised manuscript. We also added a record of C_{org} accumulation rates from our sediment core. It shows an increase during the Holocene, which is consistent with studies from shallower depths, where the primary productivity at the Peruvian ODZ decreased during the LGM (Salvatecci et al., 2016).

We added the following part to the revised manuscript (Sup. 1) and discussed the possible influence of local ODZ fluctuations:

“Local [O₂] fluctuations are directly influencing NO₃⁻ loss processes at the Peruvian ODZ. As discussed in the main text, elevated [O₂] during the LGM probably contributed to an increase in global [NO₃⁻] during the LGM. Since sediment core M77/2 52-2 is located in intermediate water depths well below the most oxygen depleted center of the ODZ a change in water column denitrification probably did not directly influence [NO₃⁻]_{BW} at this site. Benthic denitrification at depths below the Peruvian ODZ is negligible due to the lack of bacterial activity and low abundances of denitrifying foraminifera (Sommer et al. 2016; Dale et al. 2016; Glock et al. 2013), probably related to the reduced C_{org} supply compared to the shelf sediments. The decrease in water column denitrification might locally lead to an increase in [NO₃⁻] but cannot explain a decrease in δ¹³C_{DIC} which should be decoupled from denitrification. Although [O₂] was probably increased during LGM at the M77/2 52-2 site (Moffit et al., 2015, Salvatecci et al., 2016), a local decrease in denitrification cannot alone explain the tendencies observed in our sediment record, which follow to a major part the global changes in the oceanic [NO₃⁻] inventory.”

R#1: (2) preformed nitrate concentration which varies as a function of polar ocean nutrient consumption;

Reply: The issue of preformed nutrients has already been addressed above. The influence of preformed nutrients on the nitrate inventory is included in the modelling studies, discussed in our manuscript and has already been discussed in detail in the original manuscript.

Since the wording in the original manuscript might infer misunderstandings we adapted the text in the revised manuscript accordingly:

“Iron fertilization also led to additional export production and transport of remineralized NO₃⁻ into the deep Southern Ocean waters. This resulted in a reduction of preformed [NO₃⁻] in Subantarctic Mode Waters, which supply the tropical regions with preformed nutrients, affecting NO₃⁻ limitation at lower latitudes (Somes et al., 2017).

A stronger stratification of Antarctic water masses due to decreased meridional overturning during the LGM probably supported the storage of remineralized nutrients in sluggish Antarctic Bottom Water and thus supported the decrease of preformed NO₃⁻ during the LGM (Skinner et al., 2012). This led to a decreased transport of preformed NO₃⁻ to tropical ODZs limiting productivity, which restricts the extension of ODZs and thus areas of denitrification (Somes et al., 2017).”

R#1: (3) remineralized nitrate concentration which is controlled by the rate of remineralization occurring along the circulation path as well as the age of the

intermediate water. Out of these processes, (1) is strongly influenced by oxygen content, local productivity and ocean circulation.

Reply: We concede that deglacial changes in Pacific deep to intermediate water ventilation were less considered in the original manuscript. This is now discussed more in detail in the revised manuscript. See below.

R#1: While the authors argue that the different timing between their $\delta^{15}\text{N}$ change and pore density change suggest the benthic nitrate concentration is not directly related to OMZ influence, this is not a convincing argument.

Reply: This is probably a misunderstanding. We are indeed discussing the phase shift between reconstructed nitrate concentrations from the foraminiferal pore density change and the $\delta^{15}\text{N}_{\text{bulk}}$ record at the beginning of the deglaciation. We did not argue about a temporal offset between changes in these proxy records. Further arguments that benthic nitrate is probably not directly related to the local ODZ influence as discussed in sup. 1 of the revised manuscript (see above).

R#1: $\delta^{15}\text{N}$ changes speaks to the shallow OMZ where denitrification fractionation on nitrate can be reflected by surface sinking particles, so it may indeed be the case that the shallow intermediate OMZ was smaller during the last ice age. However, many evidence based on redox sensitive proxies suggests that the deep intermediate and bottom waters of the Pacific ocean was more anoxic during the last ice age, arguing that potentially there could be a shift of water column denitrification to deeper depth.

Reply: This is probably a misunderstanding too. Our coring site mirrors the intermediate water properties and their dynamics, it does not represent the deep Pacific at depths of more than 2000 m. Even there, the near-bottom waters were certainly not “anoxic”, i.e. $[\text{O}_2]$ were $<0.1 \mu\text{M}$ during the last Glacial. Otherwise, no oxic benthic foraminifera like *Cibicides wuellerstorfi* could have lived there and thereby provided stable isotope records for the respective period. Nevertheless, the contemporary literature agrees upon that the deep to intermediate Pacific was generally less ventilated during the LGM. Redox proxy records from the Eastern Equatorial Pacific and water depths of our sedimentary record showed an increased oxygenation during the LGM, though (Moffit et al., 2015). This is in good agreement with proxy records from shallower water depths of the Peruvian OMZ (Salvatore et al., 2016) and also reflected by lower C_{org} accumulation rates during the LGM at our core location, indicating decreased primary productivity (Doering et al., 2016). We are not aware of any record in the literature explicitly demonstrating enhanced denitrification at intermediate water depths of the EEP during the LGM.

R#1: But this may not be well captured by $\delta^{15}\text{N}$ changes, as the deeper denitrification signal is less well communicated with the surface ocean. At the minimum, I think this paper should look for changes of redox sensitive proxies at or near this core location.

Reply: Redox proxy records from shallower water depths of the Peruvian OMZ are already briefly discussed in the Supplement of the original manuscript. This has been extended in the revised manuscript by adding the discussion on redox proxy records from intermediate water depths of the EEP (Moffit et al., 2015). The records indicate higher oxygen concentrations at our sampling location during the LGM (Fig. R2).

Furthermore, benthic foraminiferal assemblages from our sampling location also revealed higher oxygen concentrations during the LGM (Erdem, 2016; discussion in Sup.1 of the revised manuscript). For the sake of completeness we added figure 11 from Moffit et al., 2015 to this response letter to show that intermediate water masses at the EEP showed higher [O₂] during the LGM (Fig. R2).

Figure 11. Equatorial Pacific and Humboldt Current (HC) deglacial core data synthesized into hypoxia categories. Changing deglacial core depths reflect global eustatic sea level change. The encircled number adjacent to each core label corresponds to the number of available oxygenation proxies, which are enumerated in Table 3. Vertical grey bars correlate to temporal intervals in OMZ geospatial reconstructions for this region.

Fig. R2: From Moffit et al., 2015.

R#1: (2) the preformed nitrate concentration is likely to be lower during the last ice age, so it cannot help to explain the greater nitrate concentration.

Reply: The preformed nitrate concentration at shallower water depths was indeed lower during the LGM. Nevertheless, weaker upwelling in the glacial Southern Ocean due to decreased overturning led to an increased storage of nutrients in intermediate to deep waters. This effect was even more enhanced by enhanced iron fertilization at high latitudes, increasing primary productivity and transport of nutrients to deeper waters due to the biological pump. All this has been discussed in the original manuscript but is now extended with reference to the new modelling results that are included in the revised manuscript.

R#1: However, I still need to point out that the lower ice age preformed nitrate is not because of “enhanced burial of preformed nitrate in AABW through export productivity (Line 95-96)”. Higher ice age export productivity is only observed in the subantarctic. The Antarctic export productivity is in fact lower during the ice age possibly due to stronger stratification and less wind driven upwelling.

Reply: This is probably a misunderstanding. We already discussed the decrease of upwelling in the Southern Ocean due to a more sluggish overturning. We reformulated this part of the manuscript (see above).

R#1: (3) the remineralized nitrate concentration is expected to be greater in the ice age, given the coupled oceanic carbon and nitrogen cycle, if the ocean is responsible for carbon burial, then it requires more remineralized nitrate.

Reply: Deglacial changes in the Pacific ventilation and the possible influence on remineralization are now discussed in the revised manuscript (see below).

R#1: But how much of this increase is due to whole ocean nitrate inventory change is difficult to answer.

Reply: Intriguing is, however, the striking match with model calculations on a global scale. Nevertheless, alternative scenarios are now discussed in the manuscript.

R#1: At this specific location, I would like to at least look at the ventilation age of the bottom water.

Reply: Deglacial changes in ventilation ages of the deep to intermediate Pacific are now considered in detail in the revised manuscript:

“Another factor, controlling both $\delta^{13}\text{C}_{\text{DIC}}$ and $[\text{NO}_3^-]$ in different water masses is the ventilation and thus their reservoir age. Consistent evidence from different studies indicate a poorly ventilated deep Pacific during the LGM (de la Fuente et al., 2015; Shackleton et al., 1988; Sikes et al., 2000). Data from the Eastern Equatorial Pacific (EEP) is very scarce, though, and shows some strong contrasts between different sampling locations and approaches (Broecker et al., 2004, Stott et al., 2009, Shackleton et al. 1988, de la Fuente et al., 2015). Nevertheless, it is likely that deep water masses at the EEP were also poorly ventilated during the LGM. This older water mass would increase $[\text{NO}_3^-]$ and reduce $\delta^{13}\text{C}_{\text{DIC}}$ by remineralization, as well as reduce oxygen concentration ($[\text{O}_2]$) at these depths.

Contrarily, redox proxy records from the EEP indicate higher $[\text{O}_2]$ during the LGM at depths similar to our sampling location (Moffit et al., 2015). This is consistent with other redox proxy records from shallower depths in the Peruvian upwelling region, which indicated a less pronounced ODZ and lower primary productivity during the LGM (Salvattecci et al., 2016). Indeed, the accumulation rate of organic carbon (Acc. Rate. C_{org}) at our sampling site was lower during the LGM (Doering et al., 2016) which also indicates a lower primary productivity above this sampling site (see fig. 1D). The elevated $[\text{O}_2]$ during the LGM are in disagreement with poorly ventilated water masses and thus cannot directly explain the tendencies within our record due to local changes in water mass ventilation. This suggests that local changes to overlying productivity have a strong impact on $[\text{O}_2]_{\text{BW}}$, whereas $[\text{NO}_3^-]_{\text{BW}}$ is more influenced by the global NO_3 inventory that is determined by the large-scale balance between N_2 fixation and denitrification. It might well be, though, that total changes in the nutrient budget of the Pacific are partly related to an increased reservoir age of the deep water masses, related to decreased meridional overturning.”

Answers to the comments of Reviewer #3

We thank the reviewer for the very constructive, positive and detailed feedback. We agree that our formulations were sometimes a bit too generalizing. We attempted to be more conservative with our formulations in the revised version of the manuscript. Below you can find a point-by-point response to the individual points of revision.

R#3: Reviewer #3 (Remarks to the Author):

This manuscript addresses a highly relevant topic, the reconstruction of nitrate concentrations in ancient ocean basins. This is of great importance, since nitrate plays a key role in the global nitrogen cycle and heavily influences the global carbon cycle. As such the nitrogen cycle is present in global climate models. To date, there is no faithful proxy to reconstruct past nitrate concentrations, needed for the tuning of the models.

The method presented here to estimate past nitrate concentrations, by using benthic foraminiferal pore density, is very innovative, elegant, and convincing, at least at a regional scale. It is absolutely the first time that a method capable to reconstruct nitrate concentration in oceanic bottom waters is presented. As such, the paper is of large interest for the whole paleoceanographic community, and the ideas will certainly be taken up by others. Consequently, in my opinion it should absolutely be published. The paper is fairly mature; in my opinion only a moderate revision is necessary, in which the following aspects could/should be considered:

1) The authors have a tendency to overstretch the consequences of their results. They tend to forget that their results, although being extremely valuable, pertain only to intermediate water masses in a small part of the Pacific, with a well developed OMZ. They don't have any proof that that foraminiferal pore density will be perfectly correlated with nitrate concentration in other settings, or in other ocean basins. The same holds true for the correlation between nitrate concentration and $\delta^{13}\text{C}$. I think that they should at several places be slightly more conservative in their writing.

Reply: We thank reviewer 3 for such a positive feedback and the detailed suggestions which have been made in the comments of the annotated manuscript. We agree, that the tone in the original manuscript was too generalizing in places. We followed the suggested changes remarked by the reviewer in the comments of the annotated manuscript and attempted a more conservative wording, emphasizing that the proxies have to be compared and tested in other environments. Additionally, we discussed other possible scenarios which might have influenced nitrate concentrations at our sampling location.

We added the following part into the revised manuscript to caution that the correlation might not be valid in other environmental settings or on different timescales:

"Our record and the comparison to the other deglacial $\delta^{13}\text{C}_{\text{FORAM}}$ records is considered as evidence that coupling of $[\text{NO}_3^-]$ and $\delta^{13}\text{C}_{\text{DIC}}$ was mainly controlled by the biological carbon pump at our sampling location and intermediate depths of the Eastern Equatorial Pacific, and possibly in other Pacific regions and intermediate depths. The situation might be different in the Atlantic Ocean, at greater depths or further back in Earth's history.

Independent calibrations are thus substantial to extend the application of $\delta^{13}\text{C}_{\text{Foram}}$ as a quantitative $[\text{NO}_3^-]$ proxy. Furthermore, it is unlikely that this proxy would work at shallow depths where the $p\text{CO}_2$ effect might predominate the effect of the biological carbon pump.”

Furthermore, we added the following paragraph to our conclusions:

“The biogeochemical model results of our study revealed no significant difference of the $\delta^{13}\text{C}_{\text{DIC}} - [\text{NO}_3^-]$ correlation between the LGM and the pre-industrial Holocene at intermediate water depths of the Pacific. Whereas all evidence is pinpointing that the $\delta^{13}\text{C}_{\text{DIC}} - [\text{NO}_3^-]$ correlation remained stable at Pacific intermediate depths on Glacial-Interglacial timescales, the validity of this correlation in other basins, on different time-scales or greater water depths is not yet constrained.”

R#3: 2) Related to this, the authors should try to be as precise as possible in their use of terminology and of English. For instance, $\delta^{13}\text{C}$ (an isotope ratio) can't be “enriched”. When they talk about the $\delta^{13}\text{C}$ of bottom water DIC (not foram $\delta^{13}\text{C}$), they should consistently write $\delta^{13}\text{C}_{\text{DIC}}$. More suggestions for a more precise usage of terminology are given in the annotated manuscript.

Reply: Done., also for $\delta^{15}\text{N}$ as suggested by reviewer 4 (see below).

R#3: 3) I am surprised, when the authors discuss the very consistent correlation between $\delta^{13}\text{C}_{\text{DIC}}$ and nitrate concentration in intermediate water masses, the authors don't evoke a possible control of ventilation (speed of renewal of bottom waters). In my opinion, ventilation could very well be the “the background driver controlling both parameters”. If true, this would mean that the correlation observed here is valid for intermediate waters of the whole Pacific, but the relation could be less robust in other basins, and the exact calibration will almost certainly be different. I invite the authors to reflect on this.

Reply: We agree with the reviewer that we did not sufficiently consider the possible control on nutrient remineralization by ventilation changes in the Pacific. This has also been noted by reviewer 1. Indeed, the contemporary literature agrees that deep intermediate to deep water masses of the Pacific were less ventilated during the LGM. This indeed might increase the amount of remineralized nitrate and DIC in the Pacific, leading to an increase in nitrate concentrations as well as a decrease in $\delta^{13}\text{C}_{\text{DIC}}$. Nevertheless, redox proxy records from the Eastern Equatorial Pacific and water depths of our sedimentary record show an increased oxygenation during the LGM (Moffit et al., 2015). This is in good agreement with proxy records from shallower water depths in the Peruvian OMZ (Salvatecci et al., 2016) and also reflected by lower Corg accumulation rates at our sampling location during the LGM, indicating a lower primary productivity than today (Doering et al., 2016). To address deglacial changes in Pacific ventilation ages, we added a respective paragraph to the revised version of the manuscript (see above).:

“Another factor, controlling both $\delta^{13}\text{C}_{\text{DIC}}$ and $[\text{NO}_3^-]$ in different water masses is the ventilation and thus their reservoir age. Consistent evidence from different studies indicate a poorly ventilated deep Pacific during the LGM (de la Fuente et al., 2015; Shackleton et al., 1988; Sikes et al., 2000). Data from the Eastern Equatorial Pacific (EEP) is very scarce, though, and shows some strong contrasts between different sampling locations and

approaches (Broecker et al., 2004, Stott et al., 2009, Shackleton et al. 1988, de la Fuente et al., 2015). Nevertheless, it is likely that deep water masses at the EEP were also poorly ventilated during the LGM. This older water mass would increase $[NO_3^-]$ and reduce $\delta^{13}C_{DIC}$ by remineralization, as well as reduce oxygen concentration ($[O_2]$) at these depths. Contrarily, redox proxy records from the EEP indicate higher $[O_2]$ during the LGM at depths similar to our sampling location (Moffit et al., 2015). This is consistent with other redox proxy records from shallower depths in the Peruvian upwelling region, which indicated a less pronounced ODZ and lower primary productivity during the LGM (Salvattecci et al., 2016). Indeed, the accumulation rate of organic carbon (Acc. Rate. C_{org}) at our sampling site was lower during the LGM (Doering et al., 2016) which also indicates a lower primary productivity above this sampling site (see fig. 1D). The elevated $[O_2]$ during the LGM are in disagreement with poorly ventilated water masses and thus cannot directly explain the tendencies within our record due to local changes in water mass ventilation. This suggests that local changes to overlying productivity have a strong impact on $[O_2]_{BW}$, whereas $[NO_3^-]_{BW}$ is more influenced by the global NO_3 inventory that is determined by the large-scale balance between N_2 fixation and denitrification. It might well be, though, that total changes in the nutrient budget of the Pacific are partly related to an increased reservoir age of the deep water masses, related to decreased meridional overturning.”

R#3: 4) In fact, the authors tentatively present $\delta^{13}C$ of epibenthic foraminifera as a second proxy (next to foraminiferal pore density) to reconstruct ancient nitrate concentrations in intermediate waters in the Pacific. The two proxies are totally independent. Although Fig. 1B already shows that the two downcore records (reconstructed nitrate pore density and benthic $\delta^{13}C$) are very similar, it would be interesting to compare the two nitrate proxies directly, by adding in Fig. 1C a curve with reconstructed nitrate using equation 1.

Reply: A nitrate reconstruction using equation 1 and $\delta^{13}C_{FORAM}$ of *Uvigerina peregrina* in our record has been added to figure 1C. It is following the same trend as the pore density record but showed more scatter. This is not surprising since the pore density data are averages from individual measurements on a higher number of specimens ($N \sim 20$), while the $\delta^{13}C_{FORAM}$ values are single measurements on a bulk sample of a few specimens only ($N \sim 1-5$). The following part has been added to the revised manuscript:

“The reconstructed relative $[NO_3^-]_{BW}$ changes from the pore density of *B. spissa* and another $[NO_3^-]_{BW}$ reconstruction based on the $\delta^{13}C_{FORAM}$ record on *U. peregrina* and equation 2 are showing the same trends and magnitude (Fig. 1C). The $\delta^{13}C_{FORAM}$ based reconstruction is more noisy. This is probably due to the sample size and numbers of measurements for each data point. The pore density record averages individual measurements on a higher number of specimens ($N \sim 20$) while the $\delta^{13}C_{FORAM}$ record consists only of a single measurement on a bulk sample of a few specimens ($N \sim 5$).”

Answers to the comments of Reviewer #4

We thank the reviewer for this positive and very detailed feedback. We followed every individual point of revision and think that our manuscript greatly improved due to the suggestions of the reviewer. Below you can find a point-by-point response to the individual points of revision.

R#4: Reviewer #4 (Remarks to the Author):

Review of “Coupling of oceanic carbon and nitrogen: A window to spatially resolved quantitative reconstruction of nitrate inventories” by Glock et al.

Summary

The authors use the pore-density (number of pores per unit area) in benthic forams as a proxy for bottom water NO₃⁻ concentration. From this, they recreate a record of bottom water NO₃⁻ concentration from the modern to LGM using a sediment core from the eastern Pacific (Peruvian continental margin). They then compare this record to those of $\delta^{13}\text{C}$, $\delta^{18}\text{O}$ of foram tests, with model predictions of [NO₃⁻] and with $\delta^{15}\text{N}$ of bulk organic material over the same sediment core. Among their findings – they highlight a close correspondence in the relationships between $\delta^{13}\text{C}_{\text{DIC}}$ and [NO₃⁻] at intermediate water depths in the modern ocean and that between downcore $\delta^{13}\text{C}_{\text{foram}}$ and [NO₃⁻] derived from foram pore-density measurements. Since these two relationships appear so similar – the authors propose that downcore measurements of $\delta^{13}\text{C}_{\text{foram}}$ can be used to reconstruct bottom water NO₃⁻ concentrations – and describe data that appear to corroborate this conclusion. Using this approach – the authors estimate that bottom water [NO₃⁻] in this region was ~3 μM higher during the LGM – which is also consistent with some modeling studies. The bottom water [NO₃⁻] record presented has a much higher temporal resolution than other approaches – offering unique and interesting insights into mechanisms of C and N cycling and their coupling. Two notable “side-aspects” are also included. Given the increased temporal resolution – the authors point out a brief period in which the record of bottom water [NO₃⁻] appears to diverge from modeled NO₃⁻ and offer water mass mixing effects (‘local effects’) as the primary mechanism for this. Second, the authors point out that the tight coupling between NO₃⁻ and DIC dynamics over the LGM to present interval – offers a potentially new opportunity for constraining the relative contribution of anthropogenic CO₂.

Overall, I was very impressed with the apparent fidelity of the record from pore-density measurements and with the authors’ examination of its historical record in the context of glacial/interglacial time intervals for these depths in the Pacific. This record provides a new temporal resolution for estimation of bottom water NO₃⁻ at intermediate depths and could be applied to deepen our understanding of carbon/nitrogen coupling and glacial/interglacial dynamics in a more spatially and temporally resolved manner. I would recommend publication after addressing my (mostly minor) comments below.

Major comments

1. The manuscript is very well written in general – but suffers from some English usage errors. I attempted to correct these or make suggestions in some places, but feel that the manuscript would benefit from a more thorough review for language clarity.

Reply: The revised manuscript has been scrutinized by a native speaker who is also now one of the co-authors after providing additional modelling results (Dr. Christopher Somes).

R#4: 2. The paper rests very heavily on work previously published by these authors (e.g., Glock et al., 2011). While I have no reason to doubt the veracity of these previous studies – I think that the average reader would benefit from a slightly more detailed explanation for the mechanism underlying pore-density response to nitrate – presented in the main text. While this is addressed to some degree in the Supplemental Material (Sup 4) – I feel that even just 2-4 sentences here that explain the underlying assumptions, mechanisms and limitations, would be quite useful.

Reply: A similar request has been made by Reviewer 1. We agree, a comprehensive understanding of the pore density proxy is essential for the manuscript and shifted Supplement 4 into the main text. It was also slightly extended.

R#4: Also, to this point, is it possible that this relationship is more directly linked to O₂ stress (and thus, indirectly to NO₃⁻) in the modern ocean? Where O₂ and NO₃⁻ are decoupled would this interfere in the fidelity of this proxy? For example, the relationship between $\delta^{13}\text{CDIC}$ and NO₃⁻ holds up for intermediate depths (this is well demonstrated by the presented data) – but could changes in hydrography decouple this relationship to some degree? I think that this is what is referred to in lines 106-109 and discussed in the Supplemental Material (Sup 1)? In this respect – it seems that great care is required for recognizing such shortcomings ('local effects'), and this aspect should be emphasized.

Reply: This point has been also raised by Reviewer 3. In the revised manuscript we now emphasize that the proxies have to be compared and tested in other environments. We also discussed other possible scenarios which might have influenced nitrate concentrations at our sampling location

“Our record and the comparison to the other deglacial $\delta^{13}\text{C}_{\text{FORAM}}$ records is considered as evidence that coupling of $[\text{NO}_3^-]$ and $\delta^{13}\text{C}_{\text{DIC}}$ was mainly controlled by the biological carbon pump at our sampling location and intermediate depths of the Eastern Equatorial Pacific, and possibly in other Pacific regions and intermediate depths. The situation might be different in the Atlantic Ocean, at greater depths or further back in Earth's history. Independent calibrations are thus substantial to extend the application of $\delta^{13}\text{C}_{\text{Foram}}$ as a quantitative $[\text{NO}_3^-]$ proxy. Furthermore, it is unlikely that this proxy would work at shallow depths where the $p\text{CO}_2$ effect might predominate the effect of the biological carbon pump.”

And in the outlook:

“The biogeochemical model results of our study revealed no significant difference of the $\delta^{13}\text{C}_{\text{DIC}} - [\text{NO}_3^-]$ correlation between the LGM and the pre-industrial Holocene at intermediate water depths of the Pacific. Whereas all evidence is pinpointing that the $\delta^{13}\text{C}_{\text{DIC}} - [\text{NO}_3^-]$ correlation remained stable at Pacific intermediate depths on Glacial-Interglacial timescales, the validity of this correlation in other basins, on different time-scales or greater water depths is not yet constrained.”

R#4: 3. The section discussing the coupling of $\delta^{13}\text{CDIC}$ and $[\text{NO}_3^-]$ is interesting - in particular, the possibility of related feedbacks controlling the relative proportions of N-fixation and denitrification - and implications for the global N budget. However, it was not made clear how these feedbacks (and the throttling up of benthic denitrification) would operate to modify $\delta^{15}\text{N}$, when benthic denitrification is not considered to impact the $\delta^{15}\text{N}$ of seawater NO_3^- . Perhaps I am misunderstanding something in your argument - but in either case I think there could be some attention given to improving the clarity of these arguments.

Reply: We agree and added the following sentences to the revised manuscript:

"The mean oceanic $\delta^{15}\text{N}$ is controlled by the ratio of pelagic to benthic denitrification (Galbraith et al., 2013) and the balance from N_2 fixation. The decrease of $\delta^{15}\text{N}_{\text{bulk}}$ in the Eastern Tropical North and South Pacific starting ~12 kyr BP can indeed be modeled by increasing the ratio of benthic to pelagic denitrification (Galbraith et al., 2013) since benthic denitrification fractionates $\delta^{15}\text{N}$ much less than pelagic denitrification."

R#4: 4. What is the main purpose of introducing the data from the *Cibicidoides* records at the very end? This exercise seems a little out of place in the presentation. Do you mean to demonstrate that the approach works for $\delta^{13}\text{C}$ records from carbonates in other species? Not clear.

Reply: We intended to test with the *Cibicidoides* data whether the nitrate reconstruction from $\delta^{13}\text{C}_{\text{foram}}$ using equation 1 provides similar results at other sampling locations and thus whether the downcore covariance with the pore density proxy data is not only a local phenomenon. Furthermore, we wanted to demonstrate the potential of spatially resolved nitrate concentration reconstructions based on large data sets that are readily available.

R#4: 5. It doesn't seem to me that the mid-Holocene minimum in bottom water $[\text{NO}_3^-]$ is significantly different than the other points from within the mid/late Holocene. In other words - I'm not sure it needs to be singled out as a 'feature' (L107-109; L455-458).

Reply: The interpretation of this excursion has been removed from the revised manuscript.

R#4: 6. Several important points are described by the authors in the Supplemental Material - including the influence of microhabitat preferences on foram $\delta^{13}\text{C}$, and the nature of pore-density in benthic forams, etc. While I understand the restrictions of space for publication in Nature Communications, I think it would be useful to at least briefly describe some of these points in a bit more detail - especially the topics covered in Sup 3 and Sup 4. If you are cut short on space, I think that making readers more comfortable with your assumptions of the approach might be more important than the approach for anthropogenic CO_2 (L161-165).

Reply: Supplement 4 has been moved into the main text of the revised version.

R#4: Minor comments

I would encourage the authors to use appropriate subscripts on their isotope notation for clarity. For example, the $\delta^{13}\text{C}$ values of DIC (whether measured from water or inferred from foram carbonate) would benefit throughout by a subscript (e.g., $\delta^{13}\text{C}_{\text{DIC}}$) to aid in distinguishing between $\delta^{13}\text{C}$ of foram carbonate tests and $\delta^{13}\text{C}$ of DIC.

Reply: This is done in the revised manuscript.

R#4: Similarly, I feel that the N isotopes should be explicitly denoted as referring to the bulk sediment organic matter (e.g., $\delta^{15}\text{N}_{\text{sed}}$, $\delta^{15}\text{N}_{\text{bulk}}$, or $\delta^{15}\text{N}_{\text{orgM}}$, etc.) – to make it clear that these are not somehow referring to measurements of foram bound N or NO_3^- .

Reply: Done. We are now referring to $\delta^{15}\text{N}_{\text{bulk}}$ in the revised manuscript.

R#4: Figure 1 – it would be very helpful if light lines could be added (through panels A through D) that guide the eye for the discussed time periods.

Reply: Done.

R#4: Would equation 1 be better presented flipped over and solving for past $[\text{NO}_3^-]$ here?

Reply: We added another equation (2) which is flipped over and solves for past $[\text{NO}_3^-]$.

R#4: L30: ‘is considered to the ongoing ocean deoxygenation’ – awkward – consider “is considered to be a large contributor to ongoing ocean deoxygenation.”

Reply: The sentence has been corrected in the revised manuscript.

R#4: L39: I would suggest the use of “oxygen deficient zones (ODZs)” – since oxygen minimum zones (OMZs), which occur across most water columns throughout the ocean, simply reflect elevated rates of respiration – yet may not coincide with zones of N loss via denitrification. The main point is that ODZs are regions of N loss. I realize that OMZs has been widely used to indicate the three major zones of the global ocean where N loss occurs – but feel that a more precise use of the terminology should be invoked.

Reply: Done.

R#4: L89-90: This sentence is not clear to me.

Reply: The sentence has been changed in the revised manuscript to: “A 50-100% higher reactive N-inventory is suggested for the LGM by another modeling study by Eugster et al.¹¹ and thus probably overestimates one order of magnitude of this change.” We hope that it is now better comprehensible.

R#4: L99: I find the phrase ‘burial of preformed nutrients’ somewhat peculiar. “Burial of organic matter?”

Reply: This sentence was indeed misleading and has been amended in the revised manuscript:

“A stronger stratification of Antarctic water masses due to decreased meridional overturning during the LGM probably supported the storage of remineralized nutrients in sluggish Antarctic Bottom Water and thus supported the decrease of preformed NO₃⁻ during the LGM (Skinner et al., 2010).”

R#4: L107: I think you should rephrase “ This offset and the minimum of [NO₃⁻] in our sediment record that appears during the middle Holocene from...” Otherwise this sounds like both phenomena occur at the same time – which I believe is not the intended meaning.

Reply: The discussion about the [NO₃⁻] minimum during the middle Holocene has been removed from the revised manuscript.

R#4: L116: superfluous “in”

Reply: Has been removed in the revised manuscript.

R#4: L117: Please indicate which specific time interval you are referring to here.

Reply: Time interval has been added in brackets in the revised manuscript: “(~18 kyr BP)”

R#4: L118: Either use “While” or “whereas” – but not both in the same sentence. Omit one or the other.

Reply: The sentence has been rephrased in the revised manuscript.

R#4: L120: indicates

Reply: Done.

R#4: L128: Is there a straightforward reason that δ¹³C measurements were NOT made on *B. spissa*?

Reply: Specimens of *B. spissa* were rare and they were saved for a future determination of Mn/Ca ratios as complementary redox proxy after the determination of their pore density. *B. spissa* has been shown to be a very promising archive for this proxy at the Peruvian OMZ (Glock et al. 2012). Therefore, they are not available for stable oxygen and carbon isotope measurements.

R#4: L138: established

Reply: Done.

R#4: L157: foraminifera

Reply: Done.

R#4: L169: “...do not influence δ¹³C in the same direction.”

Reply: Done.

R#4: L171: “The decreased area of continental shelves during the LGM in comparison...”

Reply: Done.

R#4: L173: I think this ‘negative’ feedback should actually refer to a ‘positive’ feedback (?). I believe you are saying that if N-fixation goes up – so does denitrification? This would be a positive feedback. If this is not the case – then please clarify.

Reply: This sentence has been removed in the revised manuscript.

R#4: L191: “Nevertheless, in order to prove coupling...”

Reply: Done.

R#4: Reference #6 is repeated as Ref #18.

Reply: The reference has been re-indexed in the revised manuscript.

R#4: Figure 1: Panel A – the $\delta^{18}\text{O}$ data is never discussed.

Reply: It is now.

R#4: Figure 1: Panel B – the green squares are very difficult to see.

Reply: It has been changed to a more visible symbol and shade of green.

R#4: Figure 1: Panel C is misaligned with the other panels.

Reply: Indeed it was it was misaligned. Thanks a lot for the remark! It has been corrected in the revised manuscript.

R#4: Figure 1: Panel D – should included error bars and refer to $\delta^{15}\text{N}$ as “ $\delta^{15}\text{N}$ of sediment organic matter.”

Reply: Done. An error bar has been added referring to 2-sigma of the reference standard (Acetanilide).

R#4: Methods:

Equation 4 – what do the small ‘deltas’ refer to?

Reply: It is a differential equation where the small delta is referring to infinitesimal small changes.

R#4: L378: I think you mean Eq 6 here?

Reply: Yes and changed.

R#4: L389: ...were included into the...

Reply: Done.

R#4: L445: ...surface water into intermediate...

Reply: Done.

R#4: L488: ...'why' there is no...

Reply: Done.

References:

Angell, R. W., The Test Structure of the Foraminifer *Rosalina floridana*, *J. Protozool.*, **14**, 299, 1967.

Bernhard, J.M. , Bowser, S.S. and Goldstein, S., An ectobiont-bearing foraminiferan, *Bolivina pacifica*, that inhabits microxic pore waters: Cell-biological and paleoceanographic insights, *Environ. Microbiol.*, **12**, 2107-2119, 2010.

Bernhard, J., *et al.*, Potential importance of physiologically diverse benthic foraminifera in sedimentary nitrate storage and respiration, *J. Geophys. Res.*, **117**, 2012.

Berthold, W.U., Ultrastructure and function of wall perforations in *Patellina corrugata* Williamson, Foraminiferida, *J. Foramin. Res.*, **6**, 22-29, 1976.

Broecker, W. S. 1974. "NO" A conservative water-mass tracer. *Earth Planet. Sci. Lett.*, **23**, 100-107.

Broecker, W. S. and T. H. Peng. 1982. *Tracers in the Sea*, Eldigio Press, Palisades, NY, 690 pp.

Broecker, W. S., T. Takahashi and T. Takahashi. 1985. Sources and flow patterns of deep ocean waters as deduced from potential temperature, salinity, and initial phosphate concentration. *J. Geophys. Res.*, **90**, 6925-6939.

Broecker, W. S. et al. Ventilation of the glacial deep Pacific Ocean. *Science* **306**, 1169-1172 (2004).

Brovkin, V., Ganopolski, A., Archer, D., and Munhoven, G.: Glacial CO₂ cycle as a succession of key physical and biogeochemical processes, *Clim. Past*, **8**, 251-264, doi:10.5194/cp-8-251-2012, 2012.

Cardich, J., *et al.*, Calcareous benthic foraminifera from the upper central Peruvian margin: control of the assemblage by pore water redox and sedimentary organic matter, *Mar. Ecol. Prog. Ser.*, **535**, 63-87, 2015.

Dale, A. W., Sommer, S., Lomnitz, U., Bourbonnais, A. and Wallmann, K. (2016) Biological nitrate transport in sediments on the Peruvian margin mitigates benthic sulfide emissions and drives pelagic N loss during stagnation events. *Deep Sea Research Part I: Oceanographic Research Papers*, 112. pp. 123-136. DOI 10.1016/j.dsr.2016.02.013.

Döring, K., Erdem, Z., Ehlert, C., Fleury, S., Frank, M. and Schneider, R. (2016) Changes in diatom productivity and upwelling intensity off Peru since the Last Glacial Maximum: Response to basin-scale atmospheric and oceanic forcing. *Paleoceanography*, 31 (10). 1453-1473 . DOI 10.1002/2016PA002936.

Erdem, Z. (2016) Reconstruction of past bottom water conditions of the Peruvian Oxygen Minimum Zone (OMZ) for the last 22,000 years and the benthic foraminiferal response to (de)oxygenation. (Doctoral thesis/PhD), Christian-Albrechts-Universität, Kiel, 2016, 201 pp.

Galbraith, E. D. et al. The acceleration of oceanic denitrification during deglacial warming, *Nature Geosciences* 6, 579-584, 2013.

Galbraith, E. D., E. Y. Kwon, D. Bianchi, M. P. Hain, and J. L. Sarmiento (2015), The impact of atmospheric $p\text{CO}_2$ on carbon isotope ratios of the atmosphere and ocean. *Global Biogeochem. Cycles*, 29, 307–324. doi: [10.1002/2014GB004929](https://doi.org/10.1002/2014GB004929).

Glock, N. *et al.* Environmental influences on the pore-density in tests of *Bolivina spissa*. *J. Foramin. Res.* **41**, 22–32, 2011.

De la Fuente, M. et al. (2015), Increased reservoir ages and poorly ventilated deep waters inferred in the glacial Eastern Equatorial Pacific. *Nature Communications*, 6, <http://dx.doi.org/10.1038/ncomms8420>.

Kaiho, K., Benthic foraminiferal dissolved-oxygen index and dissolved-oxygen levels in the modern ocean, *Geology*, **22**, 719–722, 1994.

Köhler, P. and Fischer, H.: Simulating changes in the terrestrial biosphere during the last glacial/interglacial transition, *Global Planet. Change*, 43, 33–55, 2004.

Kuhnt, T. *et al.*, Relationship between pore density in benthic foraminifera and bottom-water oxygen content, *Deep Sea Res. I*, **76**, 85-95, 2013.

Kuhnt, T. *et al.*, Automated and manual analyses of the pore density-to-oxygen-relationship in *Globobulimina turgida* (Bailey), *J. Foramin. Res.*, **44**, 5-16, 2014.

Leutenegger, S. and Hansen, H. J., Ultrastructural and radiotracer studies of pore function in foraminifera, *Mar. Biol.*, 54, 11-16, 1979.

Moffitt SE, Moffitt RA, Sauthoff W, Davis CV, Hewett K, Hill TM. Paleooceanographic insights on recent oxygen minimum zone expansion: lessons for modern oceanography. *PLoS One*. 2015;10(1) e0115246. doi:10.1371/journal.pone.0115246.

Peltier, W. R., Global glacial isostasy and the surface of the ice-age Earth: the ICE-5G (VM2) model and GRACE, *Annu. Rev. Earth Planet. Sci.* **32**, 111–149, 2004.

Petersen, C. D., Lisiecki, L. E. and Stern, J. V., Deglacial whole-ocean $\delta^{13}\text{C}$ change estimated from 480 benthic foraminiferal records, *Paleoceanography* **29**, 549-563, 2014.

Petersen, J., *et al.*: Improved methodology for measuring pore patterns in the benthic foraminiferal genus *Ammonia*, *Mar. Micropal.*, **128**, 1-13, 2016.

Pina-Ochoa, E. *et al.* Widespread occurrence of nitrate storage and denitrification among Foraminifera and Gromiida. *PNAS* **107**, 1148–1153, 2010.

Risgaard-Petersen, N. *et al.* Evidence for complete denitrification in a benthic foraminifer, *Nature* **443**, 93–96, 2006.

Salvatteci, R. *et al.* Centennial to millennial-scale changes in oxygenation and productivity in the Eastern Tropical South Pacific during the last 25,000 years, *Quat. Sci. Rev.* **131**, 102-117, 2016.

Schmittner, A., and Somes, C. J., Complementary constraints from carbon (^{13}C) and nitrogen (^{15}N) isotopes on the glacial ocean's soft-tissue biological pump. *Paleoceanography* **31**, 669–693, 2016.

Shackleton, N. J.: Carbon-13 in *Uvigerina*: Tropical rainforest history in the equatorial Pacific carbonate dissolution cycles, in: *The Fate of Fossil Fuel in the Oceans*, edited by: Andersen, N. R., and Malahoff, A., Plenum, New York, 401–427, 1977.

Shackleton, N. *et al.* Radiocarbon age of last glacial Pacific deep water. *Nature* **335**, 708–711 (1988).

Sikes, E. L., Samson, C. R., Guilderson, T. P. & Howard, W. R. Old radiocarbon ages in the southwest Pacific Ocean during the last glacial period and deglaciation. *Nature* **303**, 555–559 (2000).

Skinner, L. C. *et al.* Ventilation of the deep Southern Ocean and deglacial CO_2 rise, *Science*, **328**, 1147–1151, 2010.

Somes, C. *et al.* A Three-Dimensional Model of the Marine Nitrogen Cycle during the Last Glacial Maximum Constrained by Sedimentary Isotopes. *Front. Mar. Sci.* **4**, 108, 2017.

Sommer, S., Gier, J., Treude, T., Lomnitz, U., Dengler, M., Cardich, J. and Dale, A. W. (2016) Depletion of oxygen, nitrate and nitrite in the Peruvian oxygen minimum zone cause an imbalance of benthic nitrogen fluxes. *Deep Sea Research Part I: Oceanographic Research Papers*, **112**. pp. 113-122. DOI 10.1016/j.dsr.2016.03.001

Stott, L., Southon, J., Timmermann, A. & Koutavas, A. Radiocarbon age anomaly at intermediate water depth in the Pacific Ocean during the last deglaciation. *Paleoceanography* **24**, PA2223 (2009).

Wallmann, K., Schneider, B. & Sarntheim M. Effects of eustatic sea-level change, ocean dynamics, and nutrient utilization on atmospheric pCO_2 and seawater composition over the last 130 000 years: a model study. *Clim. Past* **12**, 339–375, 2016.

REVIEWERS' COMMENTS:

Reviewer #1 (Remarks to the Author):

I appreciate the authors added the model exercise to better constrain the uncertainties of this study and to help answer the reviewers' comments including mine. The current revision is much improved in this regard. However, I am not convinced that this current manuscript is best suited for nature communication. My main questions from last round are (1) the $\delta^{13}\text{C}$ of DIC in the ocean is not entirely driven by the same processes that change nitrate inventory; (2) this record cannot represent the global ocean. To address these concerns, the authors heavily reply on modeling exercises, which are useful to test whether some processes may or may not explain something, but cannot be used to prove something has happened in certain way. For example, when arguing the $\delta^{13}\text{C}$ of DIC did not change by changing terrestrial carbon transport, the authors used two modeling studies that suggest the depleted glacial $\delta^{13}\text{C}$ can be explained by other processes, but none of these modeling exercises prove the authors' point. To my point of view, I do not think we could use $\delta^{13}\text{C}$ to infer nitrate inventory. If it can, it is a very interesting and important topic of itself. In fact, there are more $\delta^{13}\text{C}$ reconstructions that could be used to test this idea. But this won't fit into the current format and content of this paper.

Reviewer #3 (Remarks to the Author):

The first version of the manuscript of Glock et al., entitled "Coupling of oceanic carbon and nitrogen: A window to spatially 2 resolved quantitative reconstruction of nitrate inventories" received 3 rather contrasting reviews. While reviewer 4 and I insisted on the innovative aspect and very promising character of the two new methods (using foraminiferal pore density and $\delta^{13}\text{C}$), reviewer 1 strongly criticized the claimed relationship between bottom water nitrate concentration and $\delta^{13}\text{C}$, and more in general, the treatment of the processes responsible for changes in bottom water nitrate variability. Consequently, this reviewer advised to reject the paper.

Glock et al., now present a strongly modified and expanded revised version. In my opinion, they have done a very large (and successful) effort to respond to the severe criticism of reviewer 1, and also took into account the many suggestions of the other 2 reviewers. Most important, the processes controlling bottom water nitrate concentrations, and the linkage between nitrate and $\delta^{13}\text{C}$ are explained in much more detail. The authors also added results of a modeling exercise, showing that the relationship between $\delta^{13}\text{C}$ and nitrate is stable over time, and reflects the global nitrate inventory. Another important addition is a paragraph explaining in more detail the linkage between foraminiferal porosity and bottom water nitrate concentrations, requested by two of the reviewers.

Personally, I am very satisfied with this revised manuscript, which is clearer and more persuasive than the previous version. Of course reviewer 1 is best placed to judge whether his rather severe criticism has been answered in a satisfactory way. However, even in the case reviewer 1 is not fully satisfied, I would strongly advise you to publish the paper nevertheless. This manuscript is highly innovative, and if the here proposed method to reconstruct former nitrate concentrations works, not only in intermediate waters of the eastern Pacific but on a larger scale, this would constitute a major breakthrough in paleoceanography. The claims which are made here absolutely deserve to be considered by our whole community, and to be tested on a number of different sites and time slices. As such, it seems essential to me that this manuscript is published.

All points raised in my earlier review have been treated in a very thoughtful and satisfactory way; I have no more fundamental comments, and can only indicate a couple of typos:

Line 16 – delete "deglaciation"

Line 49 – “benthos”: not clear what is meant here: “on the sea floor” ?

Line 63 – “to”  “as”

Line 74 – “reaches from”  “ranges/varies from”

Line 74 – pores may be purely ornamental. Are the authors sure of this, do they have any arguments for this statement? I would find it extremely surprising.

Line 77 – “behind”  “under”

Line 124 – “reduce”  “reduced”

Reviewer #4 (Remarks to the Author):

I feel that the authors have adequately addressed my comments and concerns - and spent substantial time and effort in addressing and revising the manuscript. While the proxy approach they have used may raise questions in other contexts - I feel that their arguments here are sound and well-presented.

Response letter to reviewers

Answers to the comments of Reviewer #1

We thank the reviewer for the effort of a second revision of this manuscript. Below you can find a point-by-point response to the review.

Reviewers' comments:

R#1: Reviewer #1 (Remarks to the Author):

I appreciate the authors added the model exercise to better constrain the uncertainties of this study and to help answer the reviewers' comments including mine. The current revision is much improved in this regard. However, I am not convinced that this current manuscript is best suited for nature communication. My main questions from last round are (1) the $\delta^{13}\text{C}$ of DIC in the ocean is not entirely driven by the same processes that change nitrate inventory; (2) this record cannot represent the global ocean. To address these concerns, the authors heavily rely on modeling exercises, which are useful to test whether some processes may or may not explain something, but cannot be used to prove something has happened in certain way. For example, when arguing the $\delta^{13}\text{C}$ of DIC did not change by changing terrestrial carbon transport, the authors used two modeling studies that suggest the depleted glacial $\delta^{13}\text{C}$ can be explained by other processes, but none of these modeling exercises prove the authors' point. To my point of view, I do not think we could use $\delta^{13}\text{C}$ to infer nitrate inventory. If it can, it is a very interesting and important topic of itself. In fact, there are more $\delta^{13}\text{C}$ reconstructions that could be used to test this idea. But this won't fit into the current format and content of this paper.

Reply: We appreciate the critical view of the reviewer about the use of $\delta^{13}\text{C}_{\text{FORAM}}$ to reconstruct past nitrate concentrations, especially on a global scale. The processes, controlling both $\delta^{13}\text{C}$ of DIC and nitrate inventories are critically discussed in detail in the revised version and the response letter from the first round of reviews. Especially the limitation to Glacial/Interglacial timescales and the intermediate Pacific, due to the lack of data in other basins or different timescales are emphasized in several parts of the manuscript. Nevertheless, due to request of the editor we added a clear caveat regarding the need for additional work to confirm the global application of the approach to the end of the discussion part (new sentence is emphasized in bold letters):

*"Whereas all evidence is pinpointing that the $\delta^{13}\text{C}_{\text{DIC}}-[\text{NO}_3^-]$ correlation remained stable at Pacific intermediate depths on Glacial-Interglacial timescales, the validity of this correlation in other basins, on different time-scales or greater water depths is not yet constrained. **We therefore caution to use this correlation on a global scale before additional work has been done on testing this proxy approach in different ocean basins.**"*

Answers to the comments of Reviewer #3

We thank the reviewer for this very positive feedback to the revised version of the manuscript. We really appreciate the effort to write such a detailed recapitulation of the

review process and evolution of this manuscript! The suggested minor revisions have been followed, which can be seen in the point by point response, below.

Reviewers' comments:

R#3: Reviewer #3 (Remarks to the Author):

The first version of the manuscript of Glock et al., entitled “Coupling of oceanic carbon and nitrogen: A window to spatially 2 resolved quantitative reconstruction of nitrate inventories” received 3 rather contrasting reviews. While reviewer 4 and I insisted on the innovative aspect and very promising character of the two new methods (using foraminiferal pore density and $\delta^{13}\text{C}$), reviewer 1 strongly criticized the claimed relationship between bottom water nitrate concentration and $\delta^{13}\text{C}$, and more in general, the treatment of the processes responsible for changes in bottom water nitrate variability. Consequently, this reviewer advised to reject the paper.

Glock et al., now present a strongly modified and expanded revised version. In my opinion, they have done a very large (and successful) effort to respond to the severe criticism of reviewer 1, and also took into account the many suggestions of the other 2 reviewers. Most important, the processes controlling bottom water nitrate concentrations, and the linkage between nitrate and $\delta^{13}\text{C}$ are explained in much more detail. The authors also added results of a modeling exercise, showing that the relationship between $\delta^{13}\text{C}$ and nitrate is stable over time, and reflects the global nitrate inventory. Another important addition is a paragraph explaining in more detail the linkage between foraminiferal porosity and bottom water nitrate concentrations, requested by two of the reviewers.

Personally, I am very satisfied with this revised manuscript, which is clearer and more persuasive than the previous version. Of course reviewer 1 is best placed to judge whether his rather severe criticism has been answered in a satisfactory way. However, even in the case reviewer 1 is not fully satisfied, I would strongly advise you to publish the paper nevertheless. This manuscript is highly innovative, and if the here proposed method to reconstruct former nitrate concentrations works, not only in intermediate waters of the eastern Pacific but on a larger scale, this would constitute a major breakthrough in paleoceanography. The claims which are made here absolutely deserve to be considered by our whole community, and to be tested on a number of different sites and time slices. As such, it seems essential to me that this manuscript is published.

All points raised in my earlier review have been treated in a very thoughtful and satisfactory way; I have no more fundamental comments, and can only indicate a couple of typos:

Line 16 – delete “deglaciation”

Reply: Done!

R#3: Line 49 – “benthos”: not clear what is meant here: “on the sea floor” ?

Reply: “benthos” has been changed to “sea-floor sediments”

R#3: Line 63 – “to”  “as”

Reply: This is probably a misunderstanding. We do not see a problem with the “to” in this line

R#3: Line 74 – “reaches from”  “ranges/varies from”

Reply: Done!

R#3: Line 74 – pores may be purely ornamental. Are the authors sure of this, do they have any arguments for this statement? I would find it extremely surprising.

Reply: The part of this sentence has now been deleted.

R#3: Line 77 – “behind”  “under”

Reply: Done!

R#3: Line 124 – “reduce”  “reduced”

Reply: Done!

Answers to the comments of Reviewer #3

We thank the reviewer for this positive feedback!

R#3: Reviewer #4 (Remarks to the Author):

I feel that the authors have adequately addressed my comments and concerns - and spent substantial time and effort in addressing and revising the manuscript. While the proxy approach they have used may raise questions in other contexts - I feel that their arguments here are sound and well-presented.